# Variance Double-Down: The Small Batch Size Anomaly in Multistep Deep Reinforcement Learning

## Abstract

In deep reinforcement learning, multi-step learning is almost unavoidable to achieve state-of-the-art performance. However, the increased variance that multi-step learning brings makes it difficult to increase the update horizon beyond relatively small numbers. In this paper, we report the counterintuitive finding that decreasing the batch size parameter improves the performance of many standard deep RL agents that use multi-step learning. It is well-known that gradient variance decreases with increasing batch sizes, so obtaining improved performance by increasing variance on two fronts is a rather surprising finding. We conduct a broad set of experiments to better understand what we call the *variance double-down* phenomenon.

## 1 Introduction

Deep reinforcement learning (DRL), which combines traditional reinforcement learning (RL) techniques with neural networks, has had a number of recent successes, including achieving superhuman performance on challenging games (Mnih et al., 2015; Schrittwieser et al., 2020; Perolat et al., 2022), overcoming difficult robotics challenges (Andrychowicz et al., 2020; Smith et al., 2022), and being successfully applied to large-scale real-world tasks (Bellemare et al., 2020; Degrave et al., 2022). Yet successful application of DRL to new problems remains a challenge, in large part due to the difficulty in understanding how neural network training is affected by the vast number of hyperparameters involved. Despite a number of recent works developing a greater understanding of the dynamics of training neural networks for reinforcement learning (Ceron & Castro, 2021; Araújo et al., 2021; Nikishin et al., 2022; Ostrovski et al., 2021; Schaul et al., 2022), the relationship between particular hyper-parameter configurations and performance on a given environment remains hard to predict.

One generally held desire in training neural networks is to reduce the variance of gradient updates, so as to avoid unstable and unreliable learning. For example, in the reinforcement learning literature there has been a growing trend to use multi-step (or $n$-step) learning (Hessel et al., 2018; Schwarzer et al., 2020; Kapturowski et al., 2018; Agarwal et al., 2022) for improved performance. Despite their demonstrated advantage, researchers have been limited to small values of $n$ to avoid performance collapse, in part due to the increased variance arising from larger $n$.

The supervised learning literature suggests that an effective mechanism for mitigating variance is through the choice of batch size: Shallue et al. (2019) empirically demonstrate that larger batch sizes result in reduced variance and increased performance. In this paper, we report the counterintuitive finding that *reducing* the batch size can help avoid performance collapse with larger $n$-step updates. This is effectively doubling down on increased variance for improved performance. We showcase this anomaly in a broad set of training regimens and value-based RL agents, and conduct an empirical analysis to develop a better understanding of its causes. Additionally, we demonstrate that reduced batch sizes also results in reduced overall computation time during training. In Appendix A we provide background on deep reinforcement learning, including a description of $n$-step updates and batch sizes.

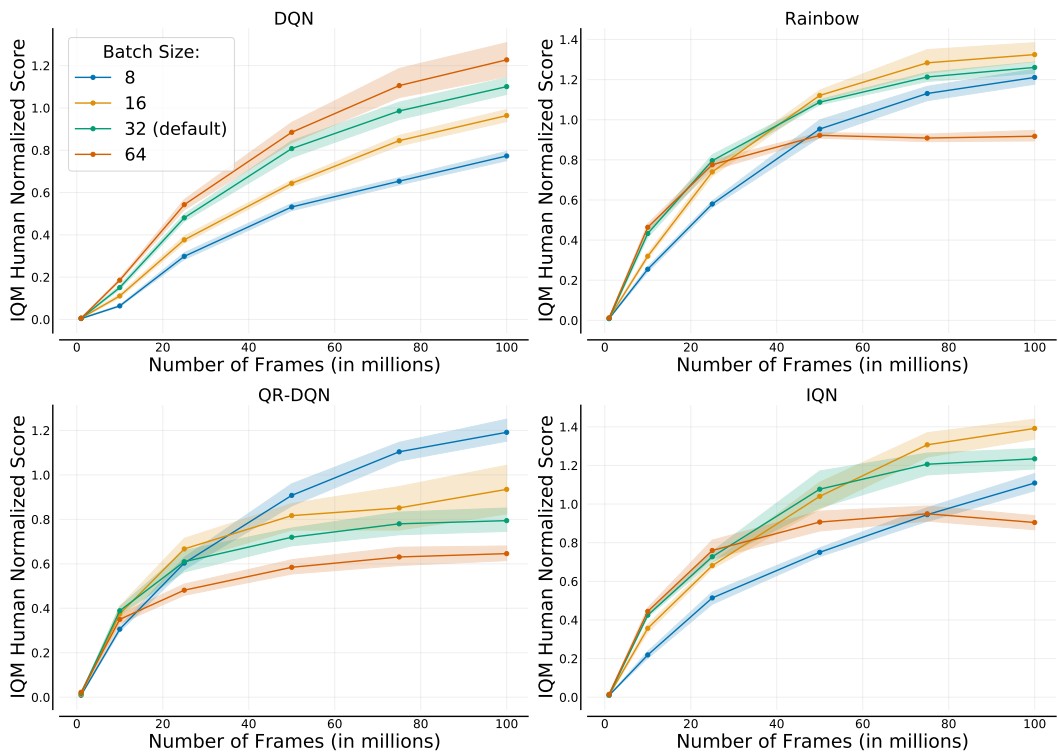

Figure 1: Varying batch sizes for DQN, Rainbow, QR-DQN, and IQN.

## 2 CASE STUDY: THE ARCADE LEARNING ENVIRONMENT

Advances in deep reinforcement learning (DRL) often build on prior algorithms, network architectures, and hyper-parameter selections. Given the large number of options, new work typically re-tunes only those components necessary for the new methods being considered. Thus, we have accumulated a set of, mostly static, parameters upon which new ideas are tested (this may be a form of the "social dynamics of research" hypothesized by Schaul et al. (2022)). One of the static parameters for training single-agent value-based agents has been the choice of batch size.

Since the introduction of DQN by Mnih et al. (2015), single-agent training on the Arcade Learning Environment (ALE, Bellemare et al., 2013) has used a batch size of 32, where this value was carefully tuned by the authors for performance. Since then, this value has rarely been changed, save for distributed agent training (Kapturowski et al., 2018; Espeholt et al., 2018). If one takes the general advice from the supervised learning literature, we should be aiming to increase the batch size so as to reduce variance and improve performance (Shallue et al., 2019). We focus on the effect of changing the batch size, while keeping all else equal.

### 2.1 EXPERIMENTAL SETUP

For this case study, we use JAX implementations of agents provided by the Dopamine library (Castro et al., 2018) and applied to game-playing in the ALE (Bellemare et al., 2013).[1] For computational reasons, we evaluate our agents on 20 games chosen by Fedus et al. (2020) in their analysis of replay ratios; these were picked to offer a diversity of difficulty and dynamics. Similarly, we run each learning trial for 100 million frames (as opposed to the standard 200 million). In exploratory experiments, we determined that for our purposes there are unsubstantial differences at 100M and 200M frames. The four agents we consider are: DQN (Mnih et al., 2015), Rainbow (Hessel et al.,

---

[1]Dopamine uses sticky actions by default (Machado et al., 2018).

2018)[2], QR-DQN (Dabney et al., 2018a), and IQN (Dabney et al., 2018b). These all use the default hyper-parameter values given in Dopamine. All experiments were run with 3 independent seeds on NVIDIA Tesla P100 GPUs.

For evaluation, we follow the robust evaluation guidelines of Agarwal et al. (2021). Specifically, we report the human-normalized *median*, *interquartile mean (IQM)*, *mean*, and *optimality gap*, aggregated over the 20 games. For all plots we report the mean with 95% stratified bootstrap confidence intervals. Agarwal et al. (2021) suggest IQM as the more robust of these metrics, so we place a stronger emphasis on it. Throughout the main paper we present the main findings, but include extra figures in the appendix.

## 2.2 WHEN DO SMALLER BATCH SIZES IMPROVE PERFORMANCE?

We first varied the batch size for all agents (Figure 1). There are two surprising observations from this result. The first is that aggregate agent performance is relatively stable with respect to changing batch sizes. The second, and perhaps more surprising, result is that *agent performance seems to improve with reduced batch size*. Indeed, we can observe that the default batch size is in fact not optimal for any of the agents and, with the exception of DQN, all agents seem to benefit from a *reduced* batch size.

The four agents considered differ in a number of respects. Three important considerations are that, of the 4, DQN is the only agent without distributional training (Bellemare et al., 2017), prioritized experience replay, and the only one without $n$-step returns. To get a better sense for whether either of these components is responsible for the reduced batch size performance boost, we performed ablation studies similar to those conducted by Ceron & Castro (2021). Since the version of Rainbow provided with the Dopamine library (Castro et al., 2018) is effectively DQN with three added components, we can investigate the changing dynamics as these components are added or removed from DQN and Rainbow, respectively. Figure 2 depicts the outcome of this ablation study. We find a striking pattern: while the four variants that use 1-step learning see their performance increase with greater batch sizes, as might be expected, the relationship is almost completely reversed for the variants using 3-step learning. Additionally, the other two components do not seem to present such a relationship with batch size. See Appendix B for further results on QR-DQN and IQN.

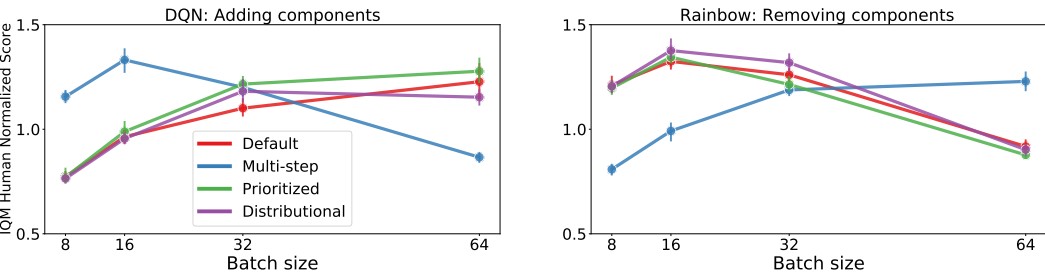

Figure 2: **Left:** Adding components to DQN; **Right:** Removing components from Rainbow.

The last results demonstrated there is a strong performance relationship between batch size and update horizon. We systematically explored this by evaluating various choices of these two parameters for three of the agents. As Figure 3 shows, the optimal batch size decreases as $n$ increases. This is most stark in QR-DQN, where simply reducing the batch size to 8 improves performance by close to 70% on the subset of games we consider.[3] With Rainbow a batch size of 8 is able to maintain performance for $n$-step values as high as 9; in contrast, performance for the default batch size of 32 collapses beyond an $n$-step of 3.

---

[2]Dopamine uses a "compact" version of the original Rainbow agent, including only $n$-step updates, prioritized replay, and categorical-distributional RL.

[3]In Dopamine, QR-DQN uses an update horizon of 3 by default.

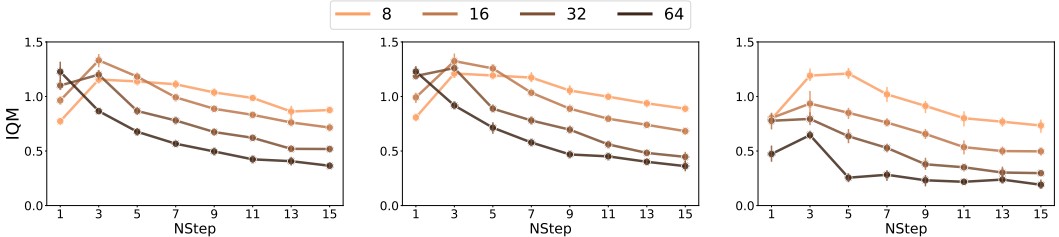

Figure 3: Varying batch sizes and $n$-steps in DQN (left), Rainbow (center), and QR-DQN (right).

## 3    UNDERSTANDING THE PHENOMENON

Having observed the advantages that can come when combining reduced batch sizes with increased update horizons, we would like to better understand the phenomenon. Given the dramatic change in performance observed in QR-DQN with an $n$-step value of 3 when reducing the batch size to 8 (Figure 3), we focus on this agent and on the game Asterix, where the effect is prominent. We provide results on additional games in Appendix C., where we observe the same qualitative findings.

### 3.1    THE EFFECT ON VARIANCE

Reducing batch size and increasing the update horizon are generally thought to *increase variance*. We thus start by confirming this through measuring the variance throughout learning.

In Figure 4 we measure, from left to right, the training returns, the variance of the TD-loss, the variance of the last layer's weight gradients, and the variance of the last layer's bias gradients.[4] The first observation is that, in aggregate, we see an increased variance with both reduced batch size and increased update horizon. Analyzing the results in more detail, we can observe the following:

**Loss variance:** As expected, this type of variance is correlated with the update horizon, but inversely correlated with batch size. In both cases, the variance seems to have an upward trend throughout training.

**Weight gradient variance:** While we see a reduction in variance with increased batch size (as expected), we seem to observe a *reduction* in variance with increased update horizon, which is the opposite of what was expected. In all cases, the variance goes down as training progresses.

**Bias gradient variance:** Here we have the expected relationship: reduced batch size and increased update horizon both bring higher variance. In contrast to the other two types of variance, this one seems to stabilize relatively early in training.

It thus appears that, in aggregate, the performance boost is correlated with increased variance on both these fronts; we are dubbing this the *variance double-down phenomenon*. The curious behaviour of the reduced variance of the weight gradients when increasing $n$-steps may be an important component of this phenomenon: perhaps it is generally advantageous to have a certain degree of variance in the weight gradients, and thus the reduced batch size helps counteract the effect of increased update horizon. Nevertheless, it is unlikely to be the only cause, so we investigate a number of additional possible causes for the phenomenon below.

**Do adaptive learning rates work better with lower batch sizes?**    All our experiments, like most modern RL agents, use the Adam optimizer (Kingma & Ba, 2015), a variant of stochastic gradient descent (SGD) that adapts its learning rate based on the first- and second-order moments of the gradients, as estimated from mini-batches used for training. It is thus possible that smaller batch sizes have a second-order effect on the learning-rate adaptation that benefits agent performance. To investigate this we evaluated, for each training step, performing multiple gradient updates on subsets of the original sampled batch; the parameter $MiniBatchSplit$ defines the number of gradient steps and dividing factor (where a value of 1 is the default setting). Thus, for a $MiniBatchSplit$ of 4, we would perform 4 gradient updates with subsets of size 8 instead of a single gradient update with the full mini-batch of size 32. With an optimizer like SGD this has no effect (as they are mathematically

---

[4]We measured the variance of the other layers but they were qualitatively the same as the final layer.

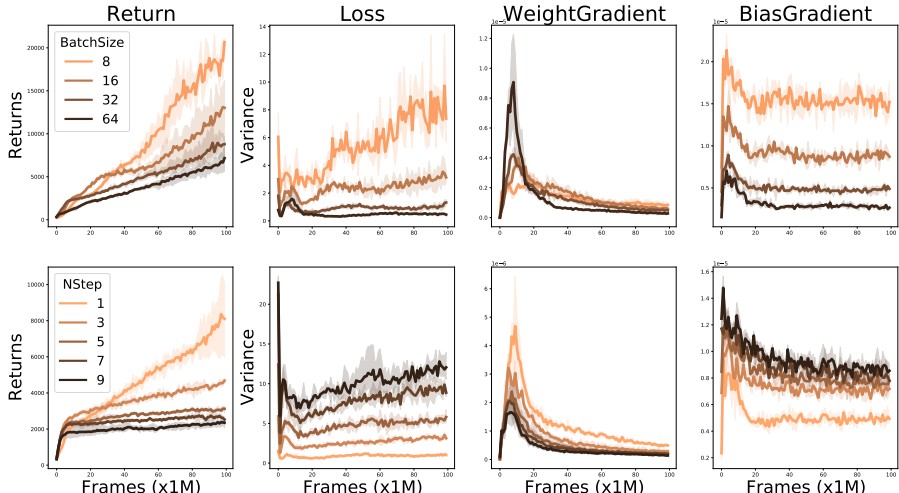

Figure 4: Measuring the variance during training while varying batch size with $n$-step value equal to 1 (top row), and varying $n$-step with batch size equal to 32 (bottom row) on Asterix.

equivalent), but we may see differing performance due to Adam's adaptive learning rates. Indeed, the left panel in Figure 5 suggests that while there are differences, these are not significant enough to explain the performance boost observed in Figure 3 when reducing the batch size to 8. We provide further analyses on all games in Appendix C.2.

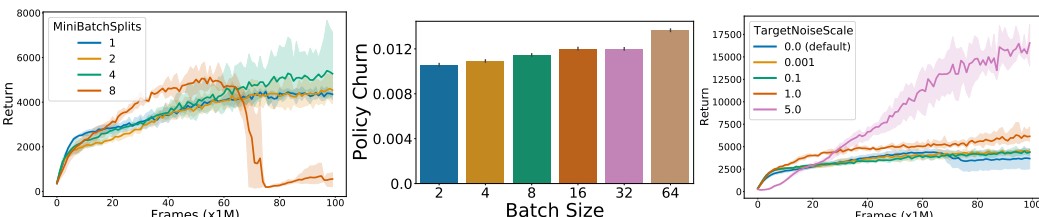

Figure 5: **Left:** Evaluating multiple gradient updates per training step; **Center:** measuring policy churn during training; **Right:** adding noise to the target update.

**Is it a side effect of policy churn?** Schaul et al. (2022) highlighted the phenomenon of *policy churn*, whereby the greedy action of a value network is rapidly and consistently changing throughout training. This effect was hypothesized to provide a training benefit in the form of *implicit exploration*. In the middle panel of Figure 5 we measured the average policy churn throughout training. We can see that policy churn increases with batch size. This is somewhat expected, as increased batch sizes effectively increases the replay ratio, which was shown to be correlated with policy churn. However, given that Schaul et al. (2022) found increased policy churn to be beneficial, it is likely that the reduced policy churn is correlated with, but not the cause of, the improved performance observed with reduced batch size.

**Is variance all one needs?** The main thesis of this work is the performance benefit obtained from the variance double-down phenomenon. Although we have focused on increasing variance in a "structured", yet indirect, way by varying update horizon and batch size, one can also increase variance in more direct ways. We explore this by adding noise to the target values used by the Bellman update during learning. Specifically, we sample a zero mean isotropic Gaussian matching the shape of the target values, scale it by a multiplier, and add it to the target values. In the right panel of Figure 5 we compare the performance when using different scaling values. Rather surprisingly, scaling the target noise by 5.0 gives a dramatic performance boost on the game Asterix. Although this striking result does not hold across all games (see Appendix C.3), it does hold for many. This

suggests that indeed, prediction variance may play an incompletely understood, yet beneficial role in deep reinforcement learning.

## 4 EFFECT UNDER DIFFERENT LEARNING REGIMES

In the preceding sections we established that learning from minibatches that are much smaller than standard results in improved performance across the gamut of Atari 2600 games. Specifically, our analysis takes place in the 100-million frames regime. At this point, one may wonder whether the root cause of the double-down phenomenon is due to peculiarities of this regime, for example the interplay between online exploration and neural network predictions (what Ostrovski et al. (2021) call the *tandem effect*). To investigate this further, we now study how the batch size parameter affects performance in other learning regimes, and how this relates to the degree to which multi-step learning is used by high-performing algorithms for this regime.

### 4.1 THE LOW DATA REGIME

We first consider algorithms designed for the *low data regime*, specifically the Atari 100k benchmark introduced by Kaiser et al. (2020).[5] This is of particular interest to us as all algorithms that achieve competitive performance on this benchmark do so by increasing the $n$ parameter beyond what is used for longer training periods. Here we consider three methods, all of which use $n = 10$: Data-efficient Rainbow (DER), a version of the Rainbow algorithm with hyper-parameters tuned for faster early learning (van Hasselt et al., 2019); SPR, which incorporates self-supervised learning to improve sample efficiency (Schwarzer et al., 2020); and DrQ($\epsilon$), which in addition uses data augmentation (Agarwal et al., 2021). Our results in this section evaluate performance on 26 games (the standard for this setting), aggregated over 6 independent trials.

Figure 6 (top) depicts the performance of all three agents after 100,000 agent steps, measured in terms of interquartile mean (IQM) of human-normalized scores (Agarwal et al., 2021). We observe that DER exhibits the same trend as the preceding experiments: reducing the batch size from its default value of 32 transitions improves performance. This is expected given that DER is a tuned version of Rainbow. The trend is less clear for SPR and DrQ($\epsilon$), although in the the former case similar performance is achieved for a smaller batch size.

These results concern published agents, whose hyper-parameters (including $n$ and, to some extent, the batch size) have been tuned to maximize performance in the 100k regime. To understand whether the double-down relationship between $n$ and the batch size parameter also holds in this regime, we evaluated these agents on a wider range of parameter values (Figure 6, middle). We find that indeed, DER exhibits this double-down relationship, with the optimal batch size varying as a function of $n$; the result is also present, to a lesser extent, with SPR and appears to be absent from DrQ($\epsilon$). This suggests that additional algorithmic components present in the latter may obviate the need for reducing the batch size in this particular regime.

Given that performance in the 100k regime depends on a number of considerations and is difficult to measure precisely, in a further experiment we trained DER and DrQ for the longer duration of 30M frames. This provides an interesting in-between the low data regime and the classic 100M regime. On this longer time frame, we indeed find that both algorithms exhibit the double-down phenomenon (Figure 6, bottom): for DER, a batch size of 8 performs substantially better, while for DrQ a batch size of 16 slightly outperforms the default value. Combined with the other results presented here, this suggests that the double-down phenomenon may only emerge with longer training regimes.

### 4.2 THE OFFLINE REGIME

We next turn our attention to the *offline reinforcement learning regime* (Gulcehre et al., 2020; Levine et al., 2020),[6] where we are given a dataset of sample transitions from which we would like to obtain a policy that performs well. Compared to the online regime, learning offline is more challenging as there is more room for overfitting to the fixed dataset, and there is no possibility for the agent to

---

[5]For comparison with the 100M frames regime presented earlier, note that "100k" in this context refers to 100,000 *agent steps*, or 400,000 *frames*.

[6]Or *batch* reinforcement learning (Ernst et al., 2005).

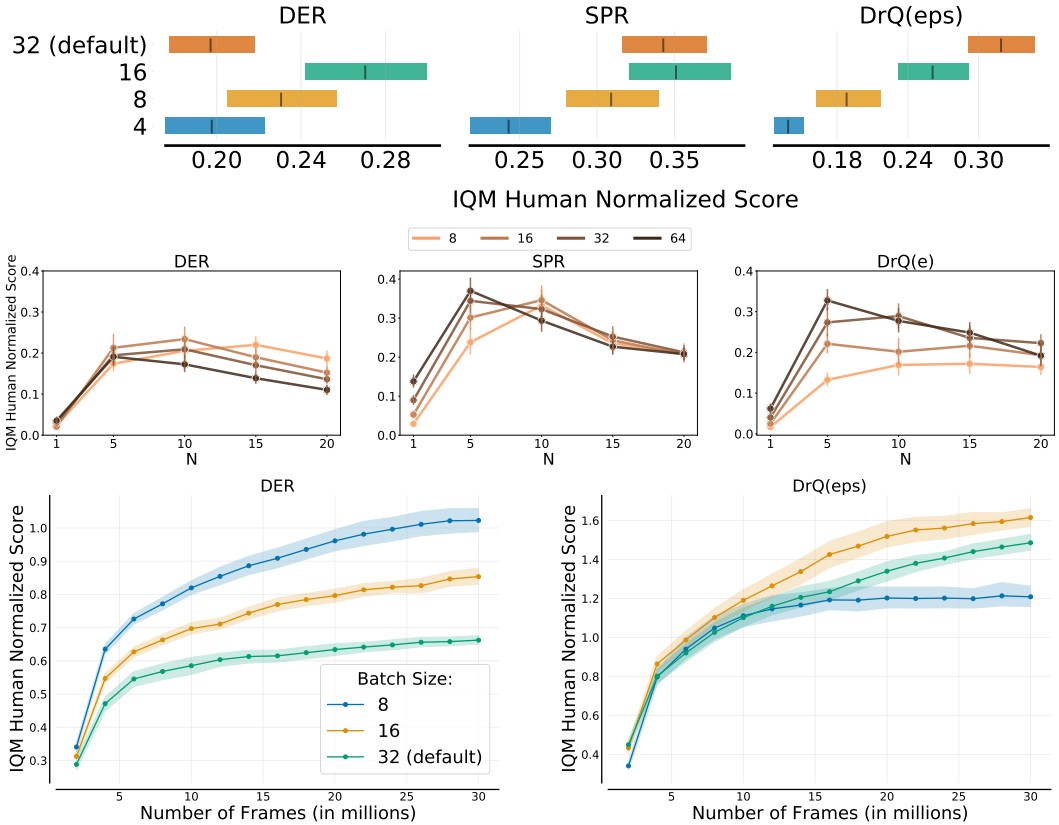

Figure 6: Varying batch sizes using the default settings of DER, SPR, and DrQ($\epsilon$), evaluated at 100k (top and middle) and 30M frames (bottom).

correct its estimation mistakes by interacting with the environment (as argued by Ostrovski et al. (2021)).

We study the effect of varying $n$ and the batch size parameter for three algorithms: DQN, CQL (Kumar et al., 2020), and CQL+DR3 (Kumar et al., 2021). Except for DQN, these algorithms are specifically tailored to the offline regime, incorporating among other things a penalty to mitigate value overestimation. We follow the training scheme of Kumar et al. (2021): each agent is trained on 17 games from the ALE for 200 iterations (where each iteration consists of 62.5K gradient steps), and after each iteration the agent is evaluated for 125K steps on the environment. The offline dataset consists of the transitions experienced during the full training of a DQN agent (Agarwal et al., 2020).

Figure 7 illustrates the impact of jointly varying our two parameters of interest on performance. In the case of CQL+DR3 (the highest-performing method), it is clear that when $n$ is increased from 1 to 3, it is beneficial to also reduce the batch size (from 32 to 4 or 8), in line with our previous findings. For CQL alone, the relative performance gap between batch sizes is reduced. We find a similar trend for DQN. One might argue that reducing the batch size without additional training effectively mitigates overestimation, simply because each transition is trained on fewer times. A closer look at the learning curves (Fig 29 in the appendix) suggests reduced overfitting is not the main factor explaining our results, at least regarding CQL and CQL+DR3.

## 5 RELATED WORK

There has been a growing interest in developing a better understanding of reinforcement learning dynamics with neural networks, and our work falls in this category. Ceron & Castro (2021) demonstrated the surprising finding that a simple switch of optimizer and loss can dramatically improve the performance of DRL agents. Andrychowicz et al. (2021) performed a broad examination of the

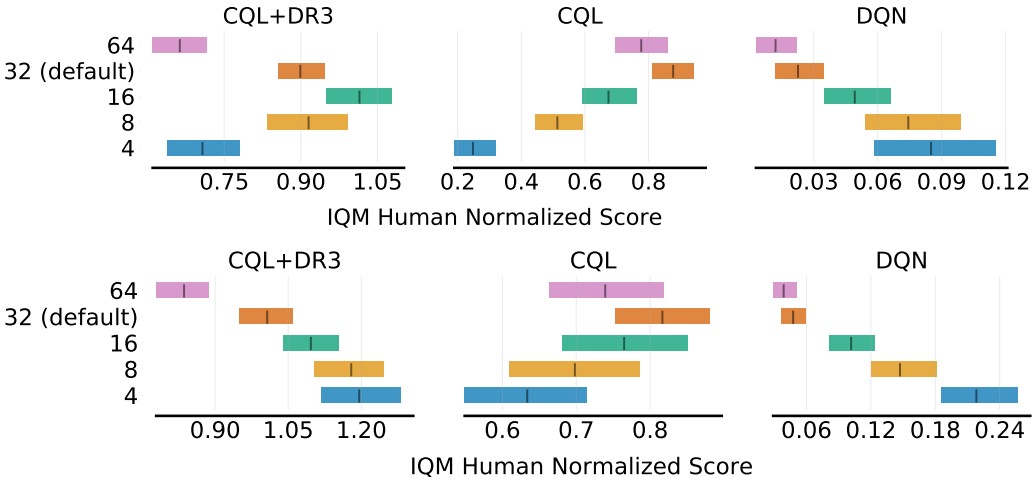

Figure 7: Varying batch sizes for offline experiments. **Top**: All agents use $n = 1$ and **Bottom**: all agents use $n = 3$.

impact varying hyper-parameters can one agent performance, for policy-gradient methods; Araújo et al. (2021) performed a similar analysis, but for value-based agents. Lyle et al. (2021) identified a mechanism by which non-stationary prediction targets can prevent learning progress in deep RL agents. They term this phenomenon as *capacity loss*. Wang et al. (2022) extensively and systematically investigated the properties of representations learned by deep reinforcement learning systems. Fujimoto et al. (2022) studied the relationship between the Bellman error and the accuracy of value functions through theoretical analysis and empirical study. They found that the Bellman error is a poor proxy for the accuracy of the value function.

The use of overparameterized deep models in value based RL still exhibits mysteries in stability and performance. To better understand the utility of deep models in RL, Xiao et al. (2022) presented an anaylsis of recursive value estimation using overparameterized linear representations. Nikishin et al. (2022) identified the primacy bias in deep RL, a damaging tendency of artificial agents to overfit early experiences. They proposed a resetting mechanism allowing the agent to forget a part of its knowledge. Fedus et al. (2020) conducted a depth study of how replay affects performance in value-based deep RL agents. Lahire et al. (2021); Stooke & Abbeel (2018) studied the key role of batch size in deep RL agents over a significant range of classical RL benchmarks.

## 6 DISCUSSION

The long-term goal of reinforcement learning research is to develop generally capable agents that can adapt to uncertain environments. Although theoretical results spanning multiple decades have given us a crisp insight into the mathematical properties of these algorithms, these theories unfortunately do not hold for non-linear function approximators such as neural networks. Given that neural networks have played a key role in the impact RL has had since 2015, it behooves the community to develop a better understanding of the the interplay of the various components and how changes can affect learning dynamics.

Our work has revealed the striking finding that *doubling down* on variance by increasing $n$ and reducing batch size seems to, overwhelmingly so, produce improved performance. This flies in the face of traditional beliefs from the supervised learning community that reduced variance is best. Indeed, the remarkable performance gains obtained from simply adding noise to the target values (right panel of Figure 5) suggest that our relationship with learning variance needs to be better understood. One important aspect of our findings is that it seems to be limited to the use of deep neural networks; indeed, in Appendix F we find that the phenomenon is *not present* when using linear function approximators.

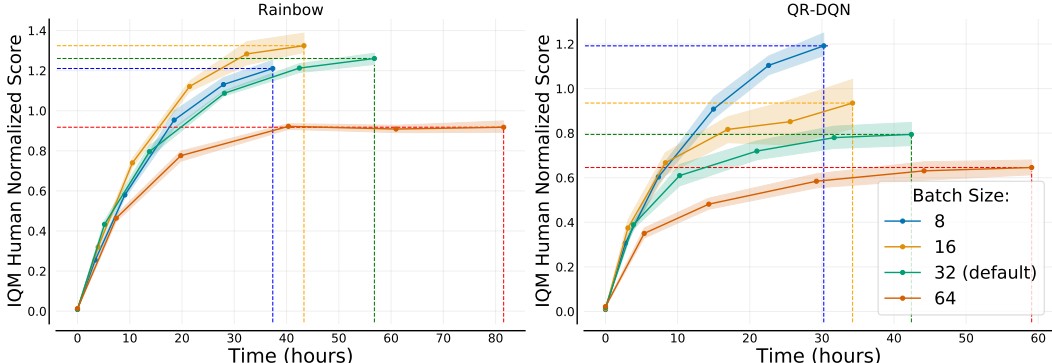

Figure 8: Measuring runtime versus performance when varying batch sizes in Rainbow and QR-DQN with $n$-step equal to 3.

One may wonder whether the same effect can be observed if one were to increase the stochasticity of action selection by, for instance, increasing the value of $\epsilon$ of the $\epsilon$-greedy strategy used by all these agents. The results in Appendix H suggest otherwise: *decreasing* $\epsilon$ seems to be more beneficial. It is interesting to observe that the variance double-down phenomenon remains present despite varying values of $\epsilon$.

One natural area for further exploration is to explore this phenomenon in actor-critic methods (Haarnoja et al., 2018; Fujimoto et al., 2018) or those agents which their return estimator is based on multi-step (Tang et al., 2022), such as Q($\lambda$) (Harutyunyan et al., 2016), Retrace (Munos et al., 2016) or emphatic algorithms (Jiang et al., 2021). We present some preliminary findings in Appendix G that suggests the variance double-down phenomenon *is* present in these scenarios. Additionally, because this paper focuses mainly on an empirical investigation of the *doubling down* phenomenon, an exciting future work it is to develop a theoretical understanding about the nature and dynamics of this phenomenon.

### 6.1 COMPUTATIONAL CONSEQUENCES

Empirical advances in deep reinforcement learning are generally measured with respect to sample efficiency; that is, the number of environment interactions required before achieving a certain level of performance (as we have done throughout this paper). While a valid metric, it fails to capture computational differences between algorithms.

If two algorithms have the same performance with respect to environment interactions, but one takes twice as long to perform each training step, one would clearly opt for the faster of the two. This important distinction, however, is largely overlooked in the standard evaluation methodologies used by the DRL community.

Many of our results have demonstrated the performance benefits obtained when reducing the batch size, but an additional important consequence is the reduction in computation wall-time. Figure 8 demonstrates that not only can we obtain better performance with reduced batch size, but we can do so at a fraction of the runtime.

We invite the reader to revisit the results presented above under this lens. For example, when evaluated with respect to environment interactions, the top row of Figure 6 would suggest that there is no real advantage to reducing the batch size from 32 to 16 for SPR; however, if evaluated with respect to computation time, the advantages of using a reduced batch would become apparent.

As argued by Ceron & Castro (2021), the ALE as a benchmark proves quite onerous for communities with limited access to compute; thus, computational gains like the one presented here can help reduce this barrier to entry. We encourage others to consider not just sample efficiency, but also computational efficiency, when evaluating new methods.

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

# A BACKGROUND

## A.1 DQN

Mnih et al. (2015) introduced DQN, which combined Q-learning with deep neural networks. Some of the most important design contributions are: (**1**) The $Q$ function is represented using a feed forward neural network consisting of three convolutional layers followed by two fully connected layers. Two copies of the $Q$-network are maintained: an *online* network (parameterized by $\theta$) and a *target* network (parameterized by $\bar{\theta}$). The online network is updated via the learning process described below, while the target network remains fixed and is synced with the online weights at less frequent (but regular) intervals. (**2**) A large *replay buffer* $D$ is maintained to store experienced transitions $(s, a, r, s')$ (Lin, 1992). (**3**) The *temporal difference update* is implemented using the following *loss function* to update the *online* network: $L(\theta) = \mathbb{E}_{(s,a,r,s') \sim U(D)}[(Y^{DQN} - Q_\theta(s,a))^2]$ using $Y^{DQN} = (r + \gamma \max_{a' \in \mathcal{A}} Q_{\bar{\theta}}(s', a'))$ and *mini-batches* of size 32 sampled from the replay buffer $D$.

## A.2 RAINBOW

In this section we briefly present the enhancements to DQN that were combined by Hessel et al. (2018) for the Rainbow agent.

Hessel et al. (2018) proposed Rainbow, a combination of several incremental improvements on top of DQN (Mnih et al., 2015) that increased sample efficiency, stability and final performance of the DQN algorithm. Rainbow rests upon years of model-free RL research and has been adapted to work well in data-limited regimes. Rainbow agent is composed of six components: double Q-learning (van Hasselt et al., 2016), prioritized experience replay (PER) (Schaul et al., 2016), dueling networks (Wang et al., 2016), multi-step learning (Sutton, 1988), noisy nets (Fortunato et al., 2018), and distributional reinforcement learning (Bellemare et al., 2017).

The two main components are (PER) and multi-step learning, removing either component caused a large drop in median performance (Hessel et al., 2018; Ceron & Castro, 2021). PER proposed to sample a trajectory $t = (s, a, r, s')$ with probability $p_t$ proportional to the temporal difference error instead of sampling uniformly from the replay buffer. In multi-step learning, instead of computing the temporal difference error using a single-step transition, one can use multi-step targets instead Sutton (1988), where for a trajectory $(s_0, a_0, r_0, s_1, a_1, \cdots)$ and update horizon $n$: $R_t^{(n)} := \sum_{k=0}^{n-1} \gamma^k r_{t+k+1}$, yielding the multi-step temporal difference: $R_t^{(n)} + \gamma^n \max_{a' \in \mathcal{A}} Q_{\bar{\theta}}(s_{t+n}, a') - Q_\theta(s_t, a_t)$.

## A.3 DISTRIBUTIONAL RL AGENTS

The experiments in the previous sections were conducted using the JAX Dopamine DQN and Rainbow implementations. However, we did not evaluate if low batch size values behave similar in Distributional Q-learning variants. In this section, we investigate the interaction of Quantile Regression for Distributional RL **QR-DQN** (Dabney et al., 2018a) and Implicit Quantile Networks **IQN** (Dabney et al., 2018b) with low batch sizes values on 20 Atari games (same setup of previous experiments).

**QR-DQN** (Dabney et al., 2018a) computes the return quantile values for $N$ fixed, uniform probabilities. This has no restrictions or bound for value, as the distribution of the random return is approximated by a uniform mixture of $N$ Diracs: $Z_\theta(x, a) := \frac{1}{N} \sum_{i=1}^{N} \delta_{\theta_i(x,a)}$, with each $\theta_i$ assigned a quantile value trained with quantile regression. **IQN** uses implicit quantile networks (IQN) as the parameterization of the return distribution Dabney et al. (2018b). IQN learns to transform a base distribution (typically a uniform distribution in $[0, 1]$) to the quantile values of the return distribution. This can result in greater representation power in comparison to QR-DQN, as well as the ability to incorporate *distortion risk measures*.

# B  THE EFFECT OF REDUCED BATCH SIZE

In this appendix we provide additional results investigating the effect of reduced batch size, complementing the results presented in section 2.

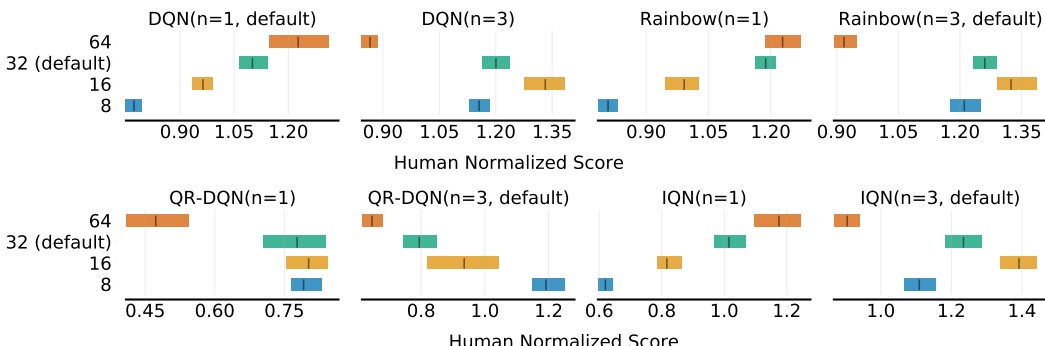

Figure 9: Evaluating different batch sizes values with two common update horizon values for DQN, Rainbow, QR-DQN and IQN.

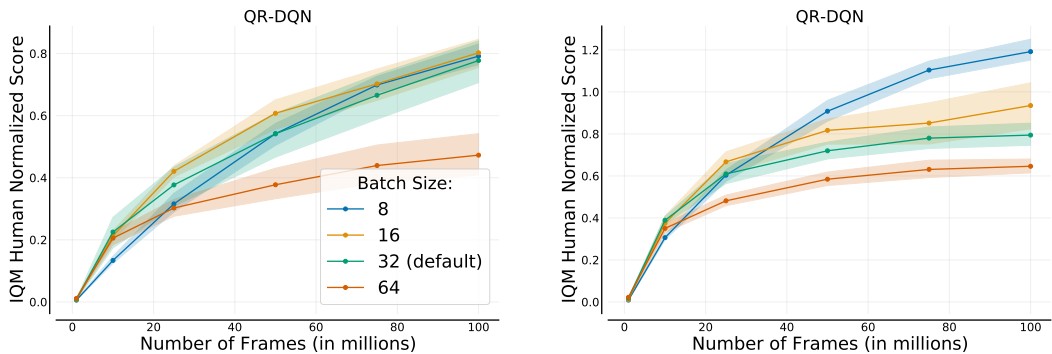

Figure 10: Evaluation of different batch size values for Quantile Regression agent with $n$-step equal to 1 (left) and 3 (right).

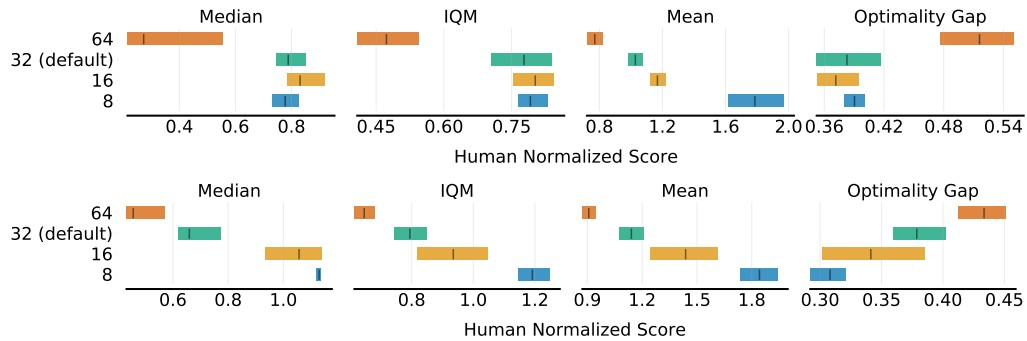

Figure 11: Evaluation of different batch size values for Quantile Regression agent with $n$-step equal to 1 (top) and 3 (bottom).

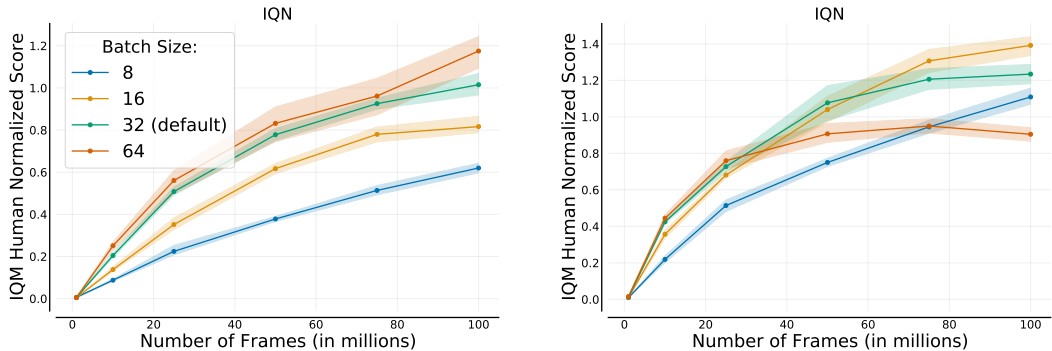

Figure 12: Evaluation of different batch size values for Implicit Quantile Regression agent with $n$-step equal to 1 (left) and 3 (right).

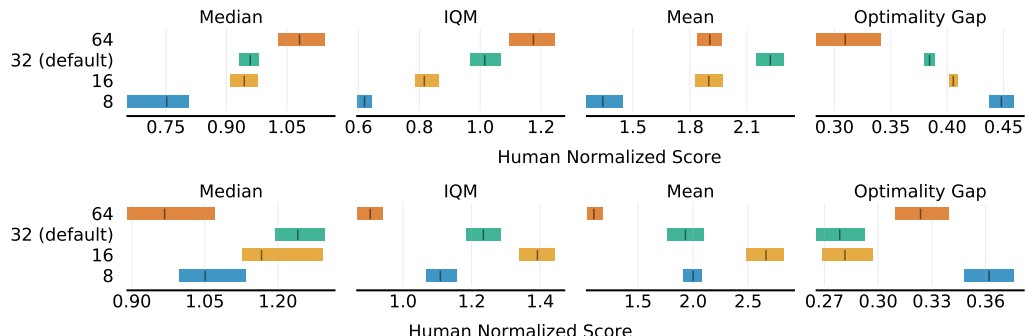

Figure 13: Evaluation of different batch size values for IQN with $n$ equal to 1 (top) and 3 (bottom).

# C  UNDERSTANDING OF THE VARIANCE DOUBLE-DOWN PHENOMENON

In this section we provide additional results to complement those presented in section 3.

## C.1  THE EFFECT ON VARIANCE

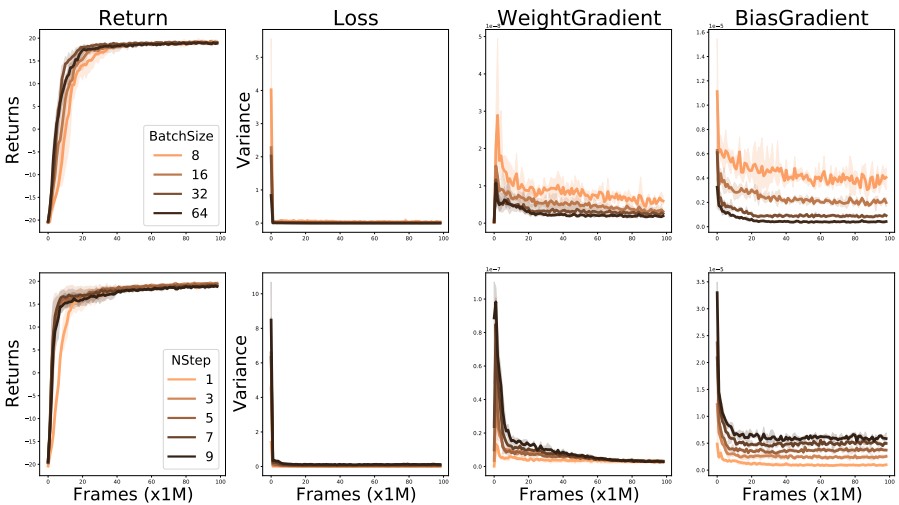

Figure 14: Measuring the variance during training while varying batch size with $n$-step value equal to 1 (top row), and varying $n$-step with batch size equal to 32 (bottom row) on Pong.

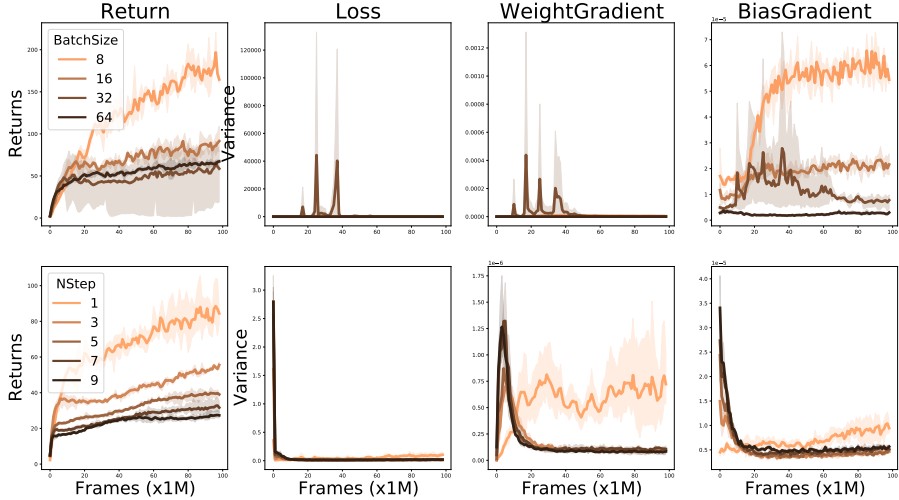

Figure 15: Measuring the variance during training while varying batch size with $n$-step value equal to 1 (top row), and varying $n$-step with batch size equal to 32 (bottom row) on Breakout.

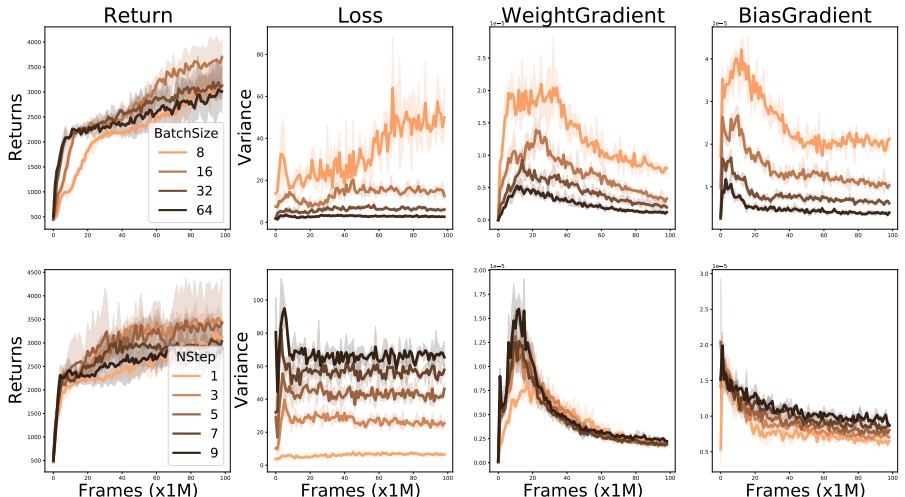

Figure 16: Measuring the variance during training while varying batch size with $n$-step value equal to 1 (top row), and varying $n$-step with batch size equal to 32 (bottom row) on MsPacman.

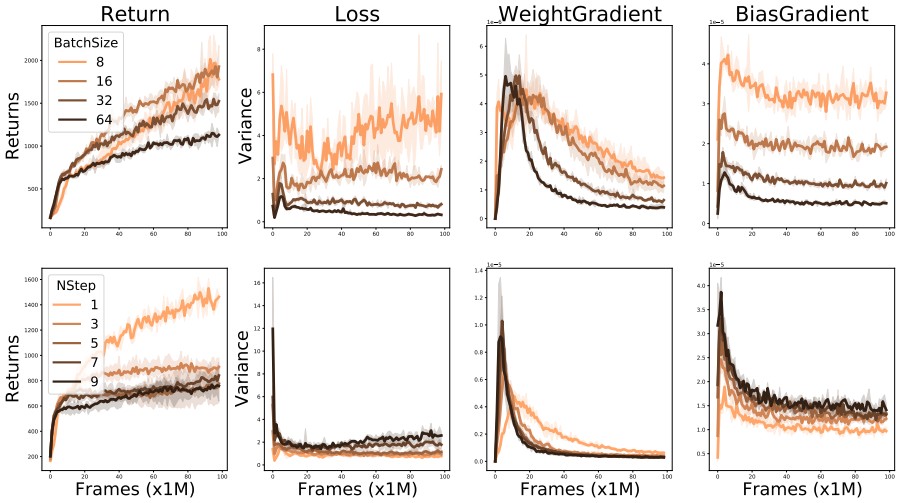

Figure 17: Measuring the variance during training while varying batch size with $n$-step value equal to 1 (top row), and varying $n$-step with batch size equal to 32 (bottom row) on SpaceInvaders.

## C.2 Do adaptive learning rates work better with lower batch sizes?

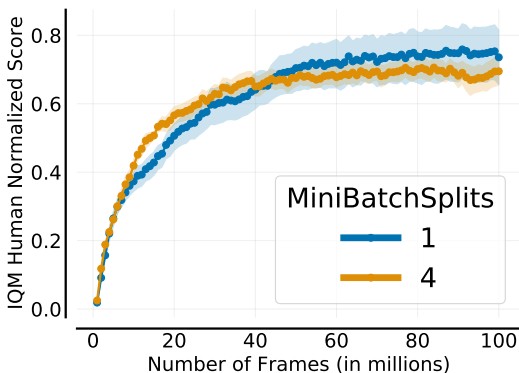

Figure 18: Evaluating multiple gradient updates per training step on QR-DQN, aggregated over all 20 games.

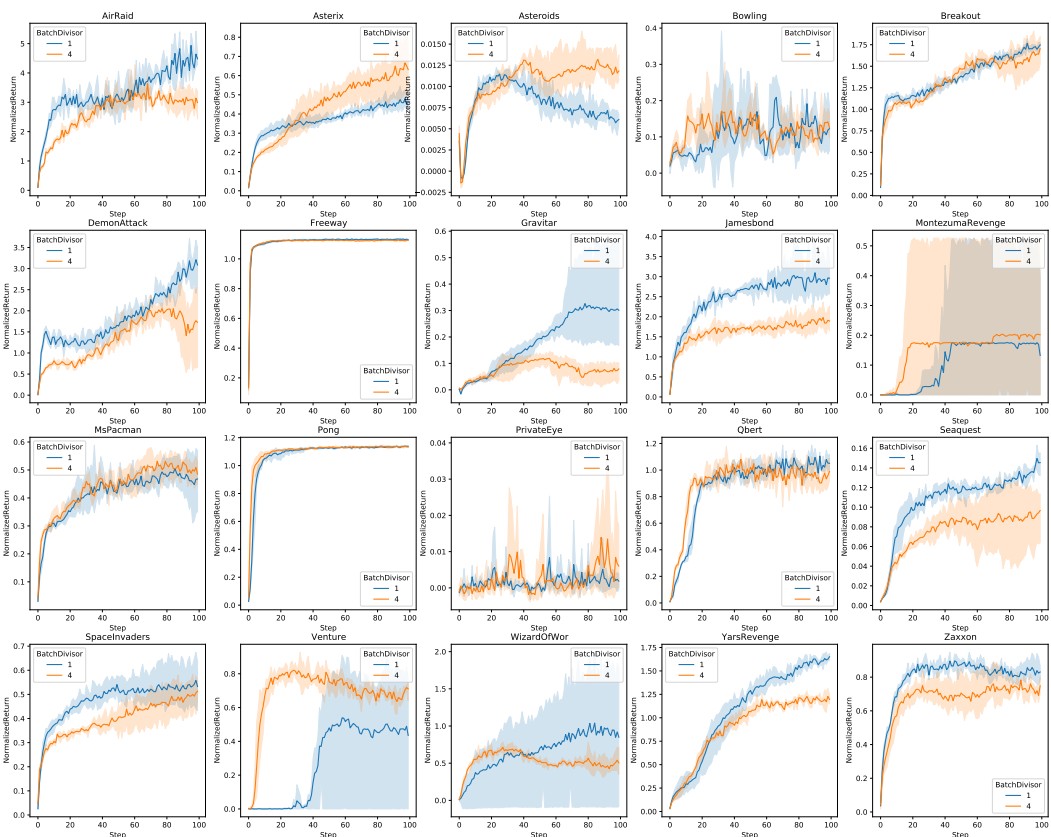

Figure 19: Evaluating multiple gradient updates per training step on QR-DQN, training curves for all games.

### C.3 Is variance all one needs?

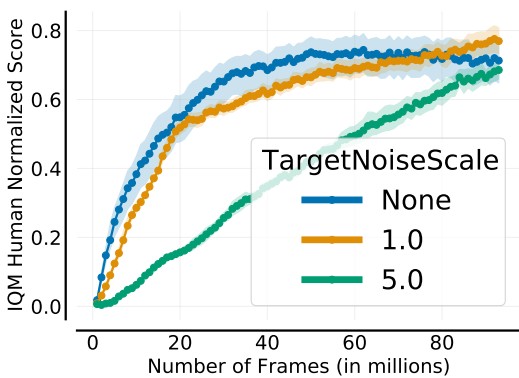

Figure 20: Evaluating the effect of adding target noise to QR-DQN, aggregated over all games.

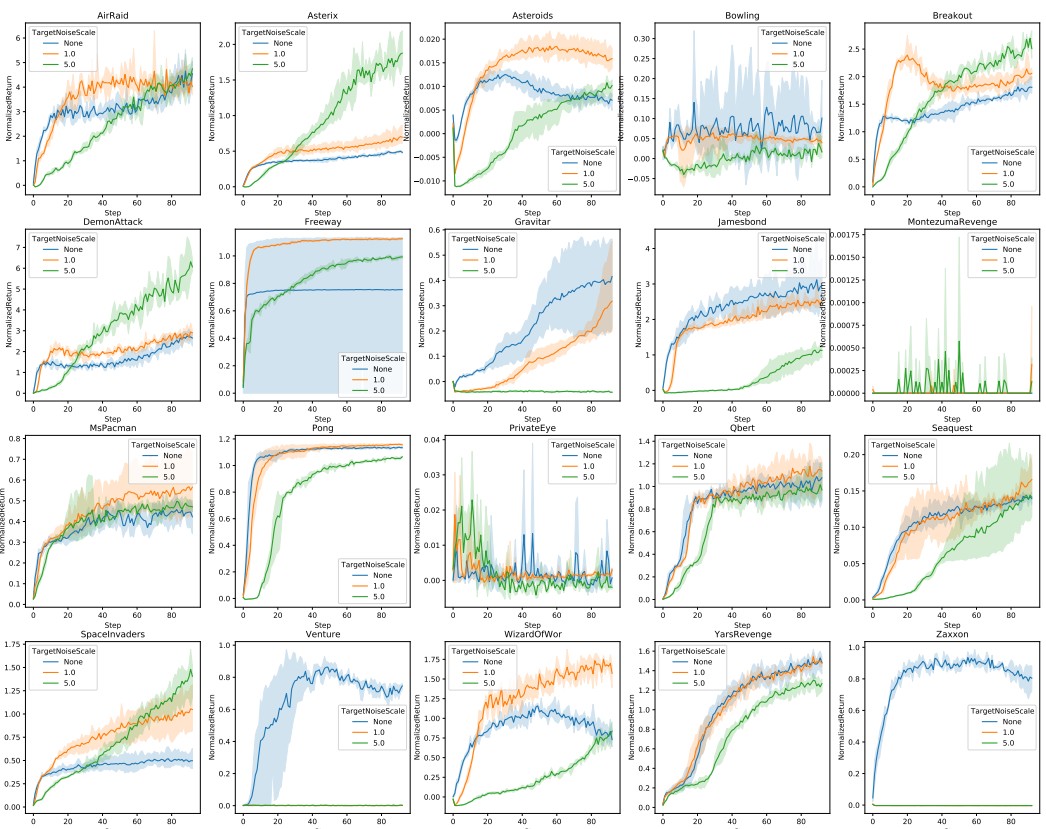

Figure 21: Evaluating the effect of adding target noise to QR-DQN, learning curves for all games.

# D  EFFECT UNDER DIFFERENT LEARNING REGIMES

In this appendix we provide additional results to complement those presented in section 4.

## D.1  LOW DATA REGIME

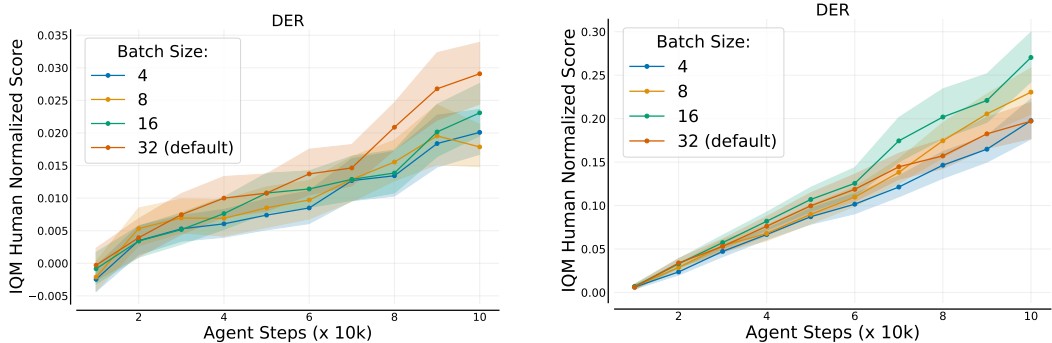

Figure 22: Varying batch sizes in DER with $n$-step equal to 1 (left) and 10 (right).

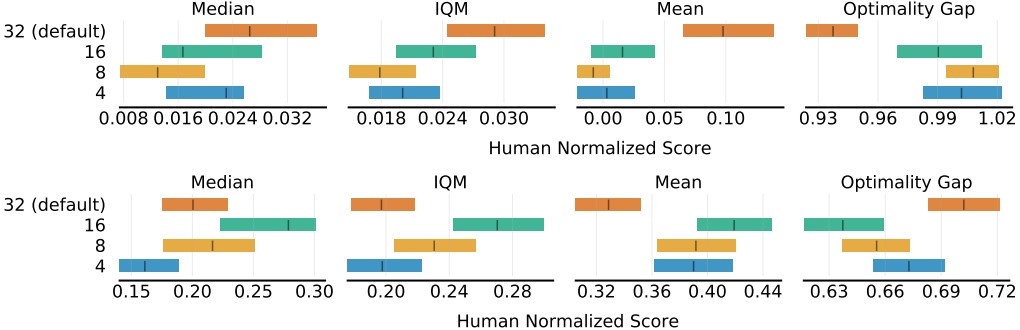

Figure 23: Aggregate metrics on Atari 100k based on 26 games. Evaluation of different batch sizes in DER with $n$-step equal to 1 (top) and 10 (bottom).

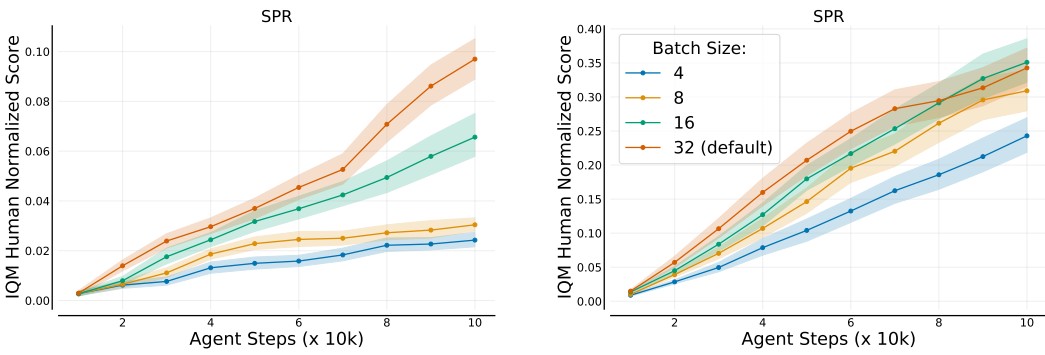

Figure 24: Varying batch sizes in SPR with $n$-step equal to 1 (left) and 10 (right)

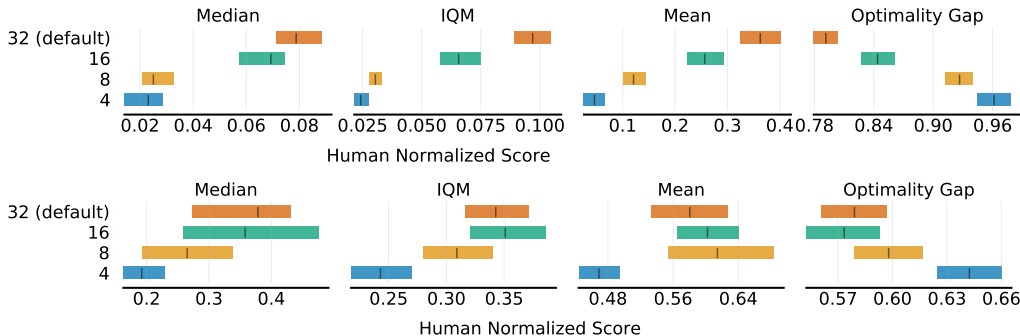

Figure 25: Aggregate metrics when varying batch sizes in SPR with $n$-step equal to 1 (top) and 10 (bottom)

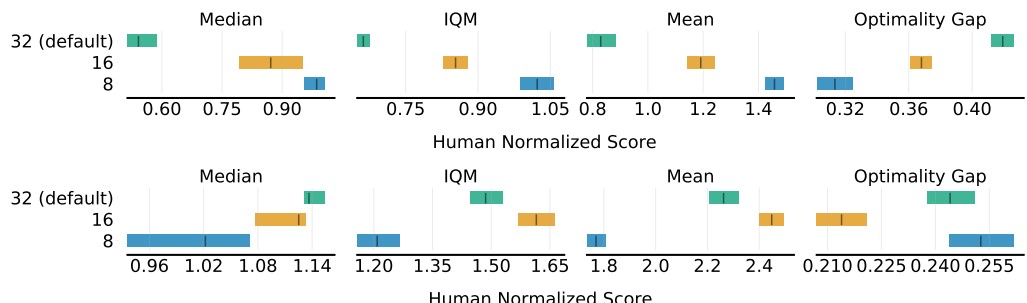

Figure 26: Aggregate metrics on Atari 30M based on 26 games. Evaluation of different Varying batch sizes for DER (top) and DrQ($\epsilon$) (bottom) using $n$-step equal to 10.

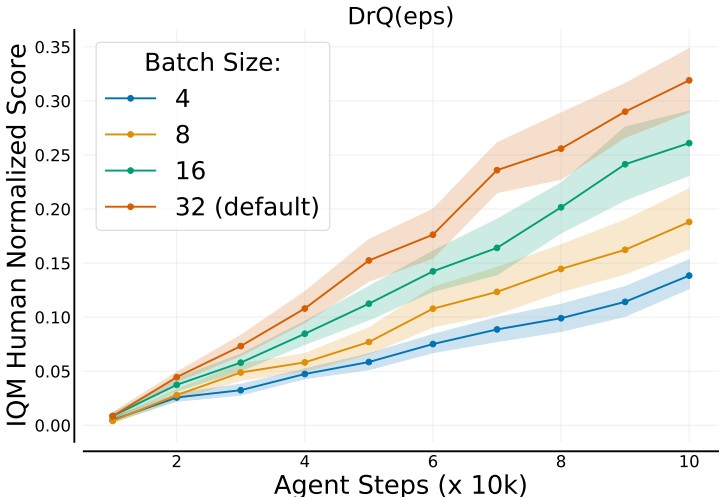

Figure 27: Varying batch sizes in DrQ($\epsilon$) with $n$-step equal to 10.

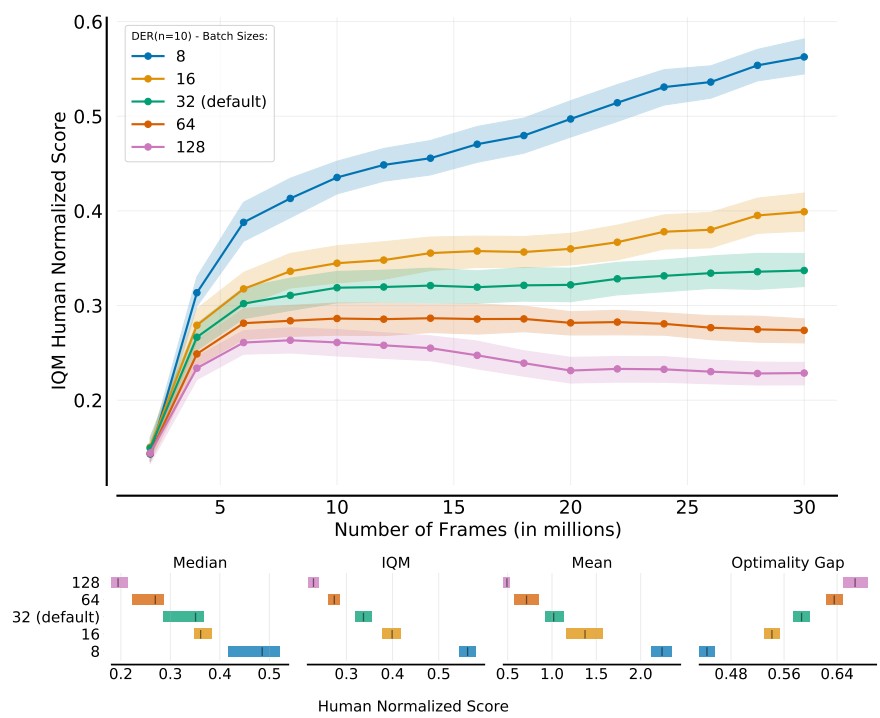

Figure 28: Evaluation of different batch size values for DER agent on Atari 30M based on the remaining 34 Atari games. **Top:** IQM curves; **Bottom:** Aggregate metrics.

## D.2   OFFLINE RL

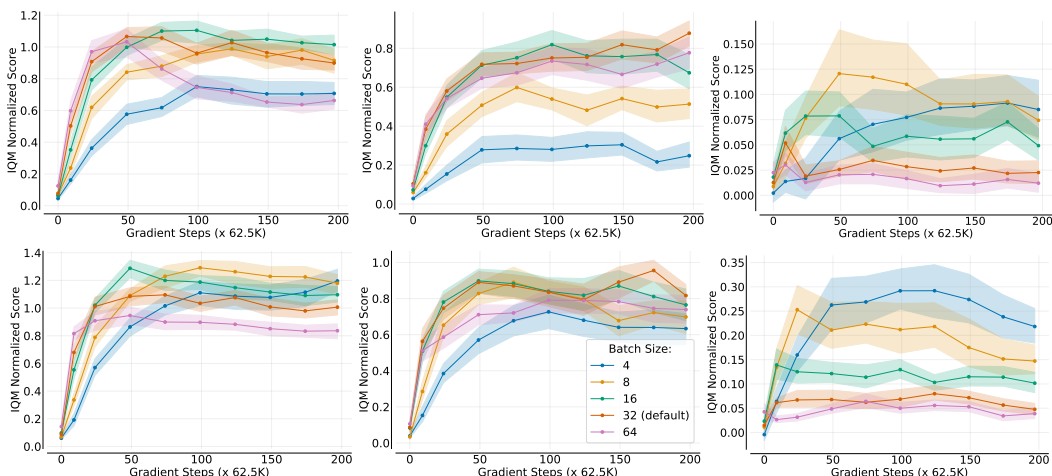

Figure 29: Varying batch sizes for CQL+DR3 (left), CQL (center), and DQN (right), with $n$-step equal to 1 (top row) and 3 (bottom row).

# E    COMPUTATIONAL CONSEQUENCES

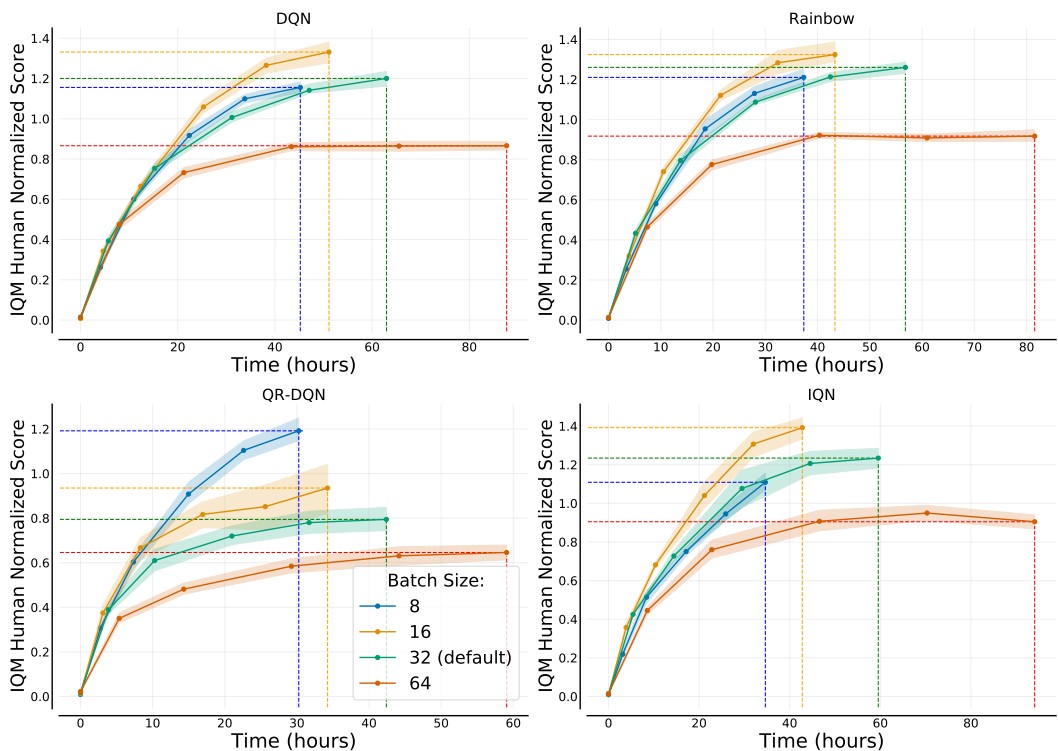

Figure 30: Measuring runtime versus performance when varying batch sizes in DQN, Rainbow, QR-DQN, and IQN (from left to right), all with $n$-step equal to 3.

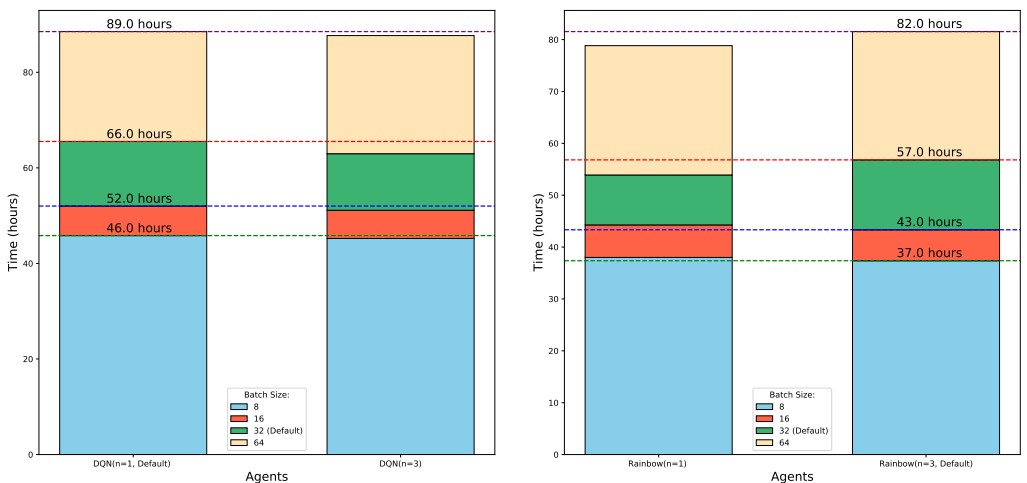

Figure 31: Computational savings: evaluating different batch sizes in DQN (left) and Rainbow (right), with $n$-step equal to 1 and $n$-step equal to 3.

# F  EVALUATION WITH LINEAR FUNCTION APPROXIMATORS

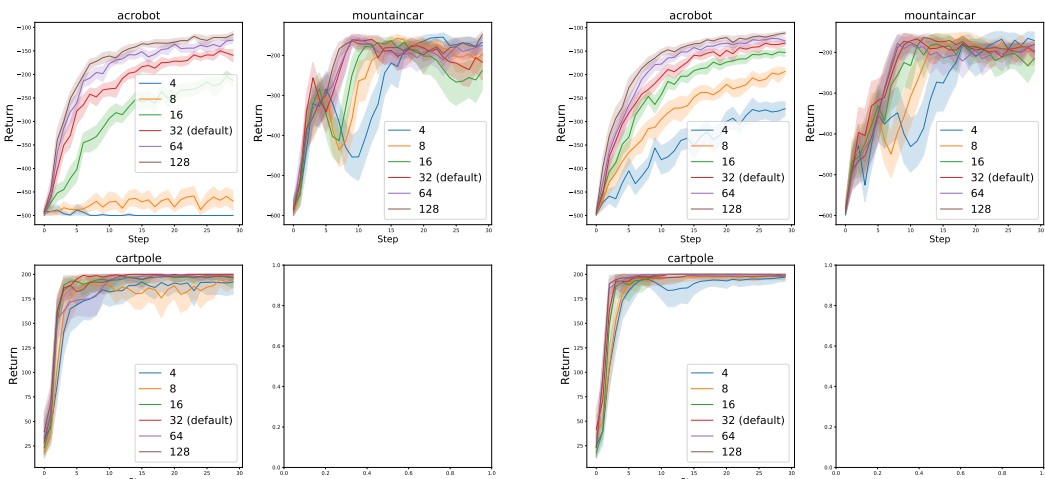

Figure 32: Evaluation of different batch size values for DQN agent using Fourier series (Konidaris et al., 2011) as a simple linear function approximation. Right: DQN($n = 1$) and Left: DQN($n = 3$).

# G IS THE PHENOMENON PRESENT IN OTHER ENVIRONMENT TYPES?

Although not as comprehensive as the results in our main paper, we found that the variance double-down phenomenon does appear to be present in continuous control tasks (Figure 33), as well as in procedurally generated environments (Figure 35).

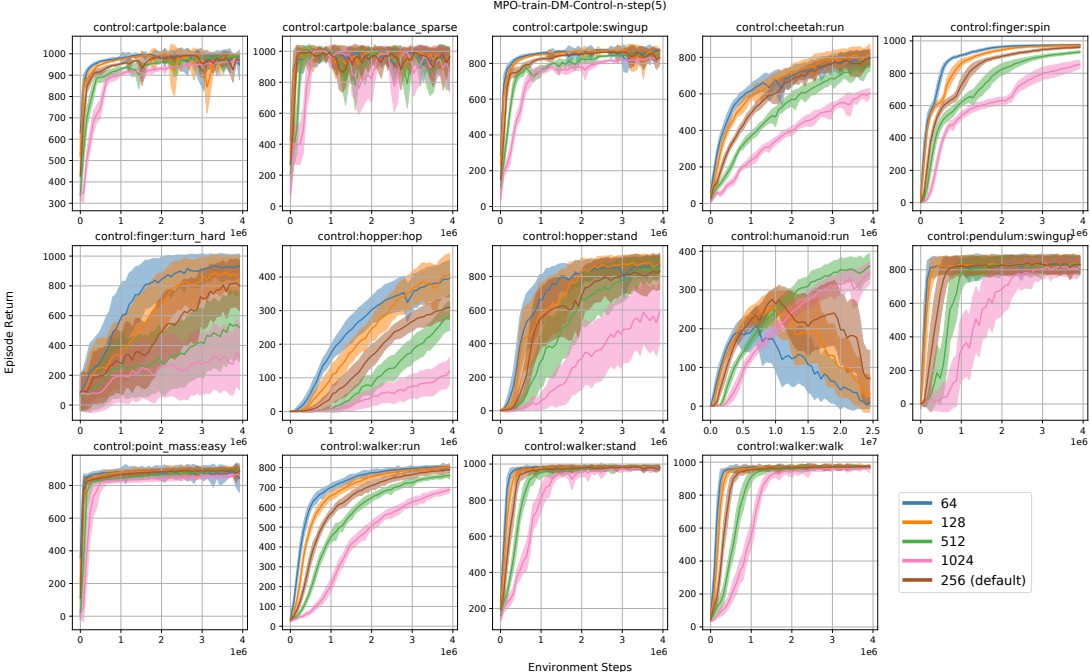

Figure 33: Evaluation of different batch size values for MPO (Abdolmaleki et al., 2018) agent on DM-control suite environments (Tassa et al., 2018), with $n$=5.

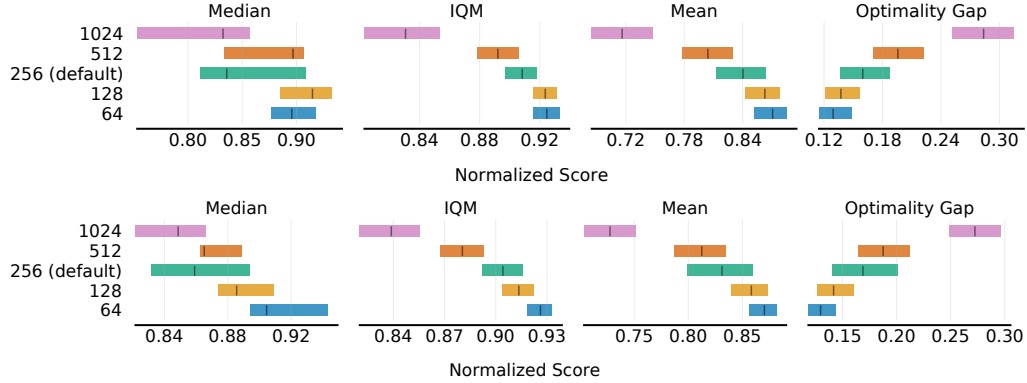

Figure 34: Varying batch sizes in MPO (Abdolmaleki et al., 2018) agent on DM-control suite environments (Tassa et al., 2018) with $n$-step equal to 1 (top) and 5 (bottom)

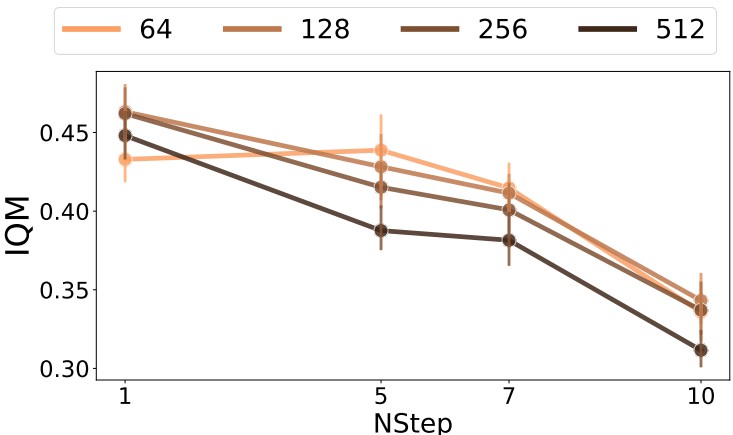

Figure 35: Evaluation of different batch size values for Rainbow agent on the ProcGen suite (Cobbe et al., 2019).

## H  EXPERIMENTS WITH VARYING $\epsilon$ EXPLORATION VALUES

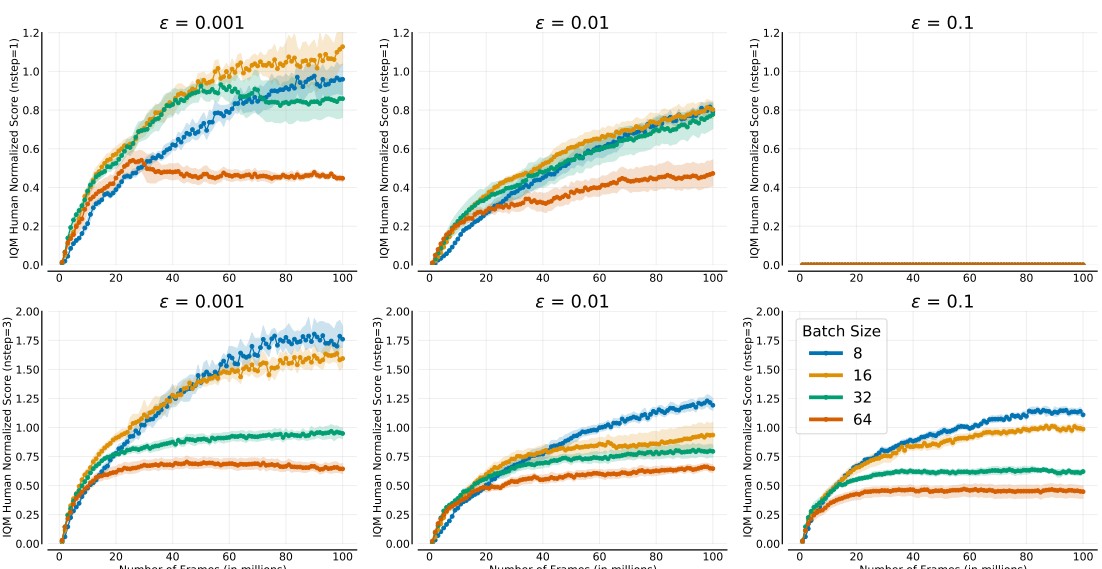

Figure 36: Evaluation of different batch size values (colored lines), $n$-steps (rows), and $\epsilon$ exploration values (columns) on QR-DQN. The default values are 32, 3, and 0.01, respectively.

# I EXPERIMENTS WITH VARYING LEARNING RATES

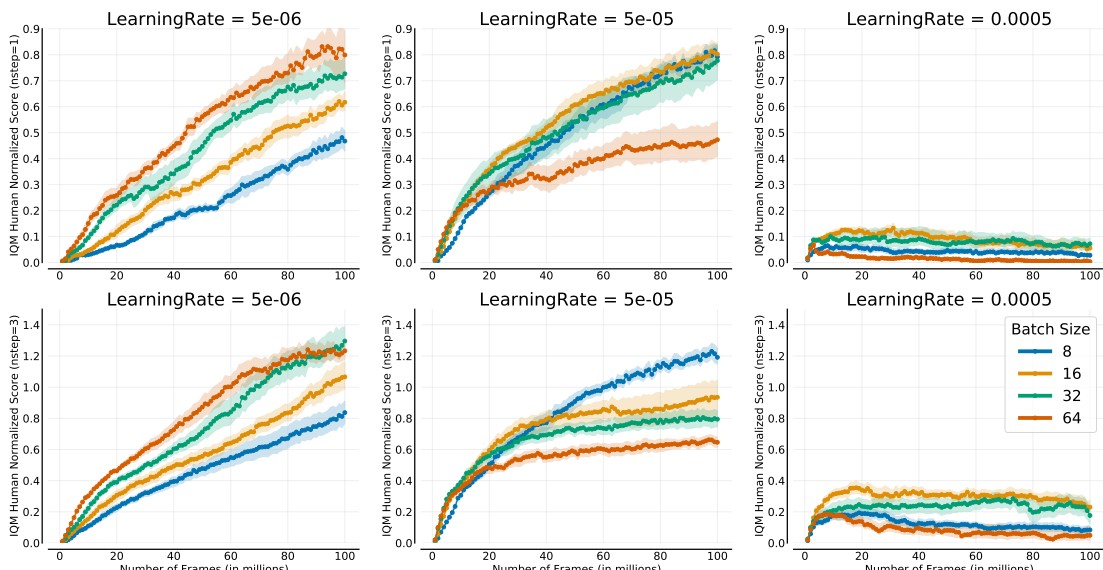

Figure 37: Evaluation of different batch size values (colored lines), $n$-steps (rows), and learning rates (columns) on QR-DQN. The default values are 32, 3, and 1e-5, respectively.

# J  ON-POLICY EVALUATION

Most of our experiments are in the online, off-policy setting, so it is informative to evaluate whether it is present in an on-policy situation. To do so, we populate the replay buffer with a fixed policy $\pi$ coming from a pre-trained agent, and do policy estimation with the same learning rule (based on the Bellman update). Figure 38 plots the $Q$-estimation error for a set of five games. The results suggest that the phenomenon is somewhat present, but only mildly so.

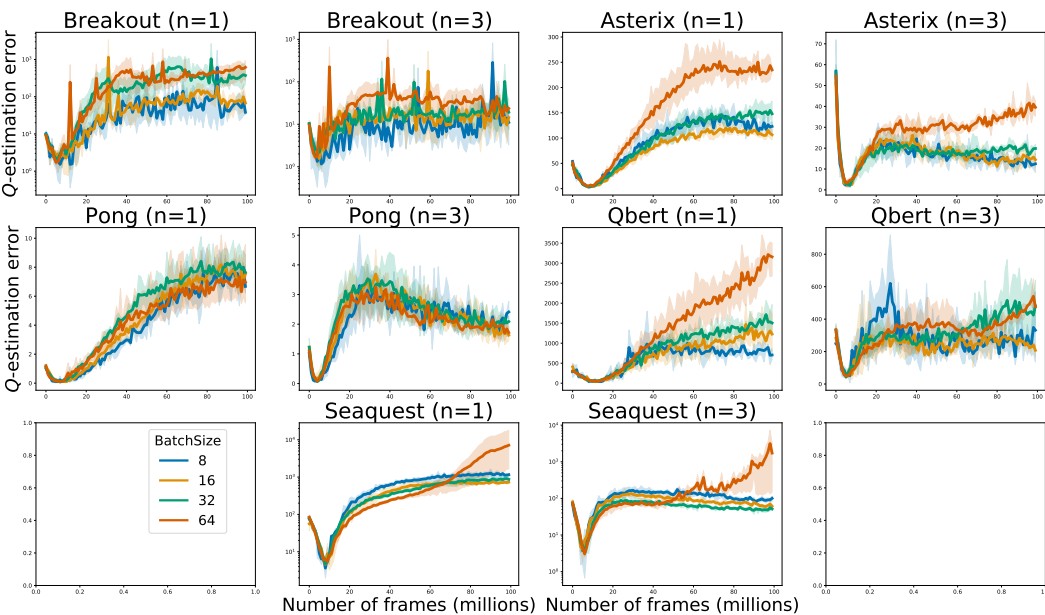

Figure 38: Evaluation of on-policy estimation errors for different batch size values (colored lines) and $n$-steps (in title) with QR-DQN.

