# OpenReview forum: "Variance Double-Down: The Small Batch Size Anomaly in Multistep Deep Reinforcement Learning"
_ICLR.cc/2023/Conference — Submitted to ICLR 2023_

### Official Review · Reviewer_iTrT · 2022-10-27

**Confidence:** 3
**Correctness:** 3
**Technical Novelty And Significance:** 3
**Empirical Novelty And Significance:** 3
**Recommendation:** 5

**Clarity, Quality, Novelty And Reproducibility:**

**Quality**

Overall in this paper, the authors studied a very interesting problem in RL; the phenomena that the authors observe is counterintuitive, which could be potentially valuable for empirical performance tuning and new perspectives of theoretical understanding of deep RL.

**Clarity**

The authors should have a deeper discussion to clarify what perspectives lead to performance improvement, which could be more valuable for future follow-up research.

**Originality**

As far as I know, there is no previous work pointing out the counterintuitive claim, which makes this work valuable.

**Strength And Weaknesses:**

**Strength**
- The authors conducted extensive experiments to show the phenomena under all kind of different settings (such as different Q-learning algorithms, DQN, Rainbow, DQN-QR, IQN). In standard Q learning settings,  the empirical experiments support the authors' counterintuitive claim.
- The paper is well written, and the experiments conducted are easy to follow.
- The claim is interesting, which is beneficial for empirical hyper-parameter selection.

**Weakness**
- It is still unclear why this claim is drawn in RL, rather than standard supervised Learning settings. There is no theoretical analysis or deeper understanding why this phenomena occurs.
- For the offline setting, why there are some counterexample that does not support your claim?
- I personally think it would be really great if the authors could point out what perspective that reducing batch size and increase the n-step in Q learning would improve the performance, especially:
  - **Exploration**. Intuitively decreasing the batch size would increase the variance for gradient updates, which potentially increase the exploration benefits.
  - **Optimization / Estimation for Q-function**. At first I think under the offline setting it would be very easy to validate whether reducing batch size and increase n in Q learning would help optimization, but from your experiments, we just see not consistent results. Besides, I think by performing an off-policy evaluation experiments (fix $\pi$ and learning Q) would check whether these two strategy would help Q estimation.
  - **Generalization**. There are previous work identifies that smaller batch size would possibly improve the performance of generalization. I wonder if your two strategy would also observe similar benefits under the RL settings.


**Summary Of The Paper:**

This paper perform an exhaustive investigation of the role of batch size and the steps used in temporal difference learning. Unlike the conventional intuition in supervised learning that increasing batch size would lead to variance reduction, thus increase the overall performance. In RL (Deep Q Learning based algorithms), decrease the batch size and increase the steps for Q-function learning would improve the performance, which is a counterintuitive finding in the literature.

**Summary Of The Review:**

Overall I think in this paper, the authors studied a very interesting question, and it is worth investigating. However, one of the major drawbacks of the paper is that we don't know why the claim is drawn in the RL settings, and also, there is no theoretical analysis or further understanding of the phenomenon.

---

> ### Author Response · Authors · 2022-11-11
> **Author response**
>
> We thank the reviewer for the thoughtful review, and we are glad the reviewer found the paper **well written, the experiments conducted are easy to follow**, and felt **the phenomena that we observe is counterintuitive**, and could be **potentially valuable for empirical performance tuning and new perspectives of theoretical understanding of deep RL**.
>
> At a high level it seems the reviewer would have liked to see this analysis conducted in a simpler setting: either in supervised learning or in a setting where theoretical analyses can be performed.
>
> As mentioned in the introduction, there is already evidence that, in supervised learning, performance gains come from larger batch sizes [1], but a similar analysis had not been conducted for DRL. This was the initial motivation for our work where we uncovered the surprising result that reducing batch size can often help. The analysis in Section 2 demonstrates that the performance gains arise when coupling reduced batch-size with multi-step updates. Multi-step updates are specific to RL, so it is impossible to evaluate the variance double-down phenomenon in a supervised learning problem.
> In the linear setting we expect theory to match known results from stochastic gradient descent for regression. In this case, reducing batch size is harmful. We added experimental results to the appendix that demonstrate that this is also the case for RL with linear function approximators, irrespective of n. Here, however, we are looking at the deep RL setting, which is known to behave differently from the linear case [2]. Unfortunately, there's no solid theoretical framework that we can build on for the deep RL setting.
>
> Detailed answers to some of the more specific concerns are below:
>
> > 1) For the offline setting, why there are some counterexample that does not support your claim?
>
> The variance double-down phenomenon is clearly visible in 2 / 3  of the agents considered (DQN and CQL+DR3). The fact that the effect is only mildly evident  in CQL may be a consequence of the  implicit under-parameterization exhibited by this algorithm (and which DR3 aimed to mitigate).
>
> > 2) I personally think it would be really great if the authors could point out what perspective that reducing batch size and increase the n-step in Q learning would improve the performance, especially: exploration:
>
> As stated in our response to R2, we are running some extra experiments varying the target epsilon value to ensure the uncovered phenomenon is unaffected. Although the agents have not completed training, the results suggest decreasing $\epsilon$ seems to be more beneficial. It is interesting to observe that the variance double-down phenomenon remains present despite varying $\epsilon$ values. We have added these results to Appendix H, and included a discussion in Section 6.
>
> **Optimization / Estimation for Q-function:** We thank the reviewer for this suggested analysis (fixing π and learning Q with varying batch-sizes/nsteps). We will run these experiments as soon as the current ones are finished running, due to computational constraints. We will report back here when they are done.
>
> **Generalization:** We assume the reviewer is referring to supervised learning when mentioning previous work? There is no universally adopted  definition for “generalization” in RL, since agents typically train on the data they themselves generated. However, the offline setting does allow us to test this to some extent: the agent is training on data that it did not generate, and is then evaluated on the real environment. The results in the offline setting thus suggest that perhaps there is some gain in generalization, although this requires further investigation.
>
> > 3) The authors should have a deeper discussion to clarify what perspectives lead to performance improvement, which could be more valuable for future follow-up research.
>
> The analyses performed in section 3 were aimed at this, which suggest the performance increases are a consequence of increased variance from reduced batch-size and multi-step learning. However, we will extend our discussions with some of the connections suggested by the reviewers.
>
>
>
> ### References:
> 1. Shallue, C. J., Lee, J., Antognini, J., Sohl-Dickstein, J., Frostig, R., & Dahl, G. E. (2018). Measuring the Effects of Data Parallelism on Neural Network Training. doi:10.48550/ARXIV.1811.03600
> 1. C. Lyle, M. G. Bellemare, and P. S. Castro, “A Comparative Analysis of Expected and Distributional Reinforcement Learning”, AAAI, vol. 33, no. 01, pp. 4504-4511, Jul. 2019.
> 1. G. Konidaris, S. Osentoski, and P. Thomas, “Value Function Approximation in Reinforcement Learning Using the Fourier Basis”, AAAI, vol. 25, no. 1, pp. 380-385, Aug. 2011.

---

> ### Author Response · Authors · 2022-11-18
> **Fixing π and learning Q with varying batch-sizes/nsteps**
>
> We have included the results of the proposed experiment (fixing $\pi$ and learning $Q$ with varying batch-sizes/nsteps) in Appendix J. The results suggest that the phenomenon is somewhat present, but only mildly so. As such, it appears to be a phenomenon that is mostly present in the online, off-policy, training regime. We will add discussion of this point in the final version.

---

### Official Review · Reviewer_74JP · 2022-11-01

**Confidence:** 3
**Correctness:** 2
**Technical Novelty And Significance:** 1
**Empirical Novelty And Significance:** 2
**Recommendation:** 3

**Clarity, Quality, Novelty And Reproducibility:**

**Clarity** : This study gives experimental findings on the ALE framework. If the authors wish to assert that a small number of batches increases performance by increasing stochasticity, they should examine more environment like Mujoco. If not, at least, the ALE-like Procgen should be chosen to support this notion. As just one setting employs ALE, its utilization is marginal.

**Quality** : This document appears to lack a logical structure. In addition, this study does not present the state-of-the-art performance on ALE, nor does it present consistent empirical evidence for substantially direct performance increase. Simply said, adjusting mini-batches has a minor effect on performance, however not all algorithms adhere to this hypothesis. If the authors give an examination of algorithms by modifying the batch size in RL, they must develop more high-level technical statements that enable comprehension of the exploration and exploitation balance, since this is a fundamental foundation for comprehending decision making areas. Otherwise, they should at least characterize the entropy of of batches for learners during training.

**Novelty** : The authors discover that the batch size of RL has a positive impact on performance. Previously, the majority of RL articles focused on network architectures and manipulation, therefore this discovery is expected to give new axes for improving RL algorithms. However, as previously said, I felt that this research lacked sufficient results to conclude these finding in ICLR2023.

**Reproducibility** : In the manuscript, the authors described all algorithms follow the Dopamine framework, however, I believe that it did not meet the ICLR standard for reproducting experimental results. The authors do not provide ant source-codes and checkpoints to validate this results. Specifically, due to the fact that this paper employs empirical way to reasoning, I though that providing source-code and checkpoints was crucial, however it does not.

**Strength And Weaknesses:**

### Strength

- Numerous experimental findings altering hyper-parameters such as batch sizes and step-sizes of multi-step value functions are presented in this study.
- This article shown that ablation research varies each component of deep value-based RL.
- By changing the batch size, they have repeatedly shown that agents get higher results in the ALE framework.
- Motivated by the stochasticity of networks in supervised learning, they studied the gradient values of networks based on the number of batch-sizes.

### Weakness

- This study provides neither a technical nor a theoretical explanation of batch size and multi-step in deep reinforcement learning. I was unable of digesting a high-level summary of this research. Specifically, I disagreed with the author's statement that multi-step reinforcement learning should be examined independently. Numerous empirical research have already demonstrated that the multi-level value function does not exhibit a consistent trend across diverse situations. As the authors noted out in their paper, past research has, as a rule of thumb, set the value of multi-level performance to three. At this time, I was unable to understand why this work manipulates and reveals the effect of multi-levels and its impact on batch-size by up to 15.

- Numerous theoretical works [1, 2, 3] have examined how the stochasticity of gradients is obtained from batch components. These studies also asserted that random mini-batches (shuffling) demonstrate networks that are more generalized than static mini-batches. I guess that these research are incompatible with online learning, but I believe they might give insightful perspectives on the exploration of batch size for off-line learning. In particular, in theoretical RL, the majority of publications might be demonstrating that such an algorithm can create a transitions matrix from provided datasets that include batches. At least at this stage, I believe the authors should provide a high-level statement regarding batch size to grasp the stochasticity of learners rather than give basic experiments to prove their hypothesis.

- There are several hyper-parameters in RL, such as $\epsilon$-greedy, learning rate, and update frequency. However, this article does not consider the exploration impact in online learning. Exploration and exploitation must be balanced very carefully during the decision-making research. However, this article only discussed the stochasticity of networks and its association to performance; there is no investigation of the exploration impact. To validate your hypothesis, you must offer consistent experimental results with modifying these hyper-parameters related to exploration.

- The experimental setting was inconsistent in this work. In Figure 1, DQN, Rainbow, and IQN were employed as baselines, but in Figure 2, just DQN and Rainbow were used for the ablation investigation. Specifically, I do not understand why the mini-batch split and random noise to target-value experiment results were reported. I felt that these trials were not really relevant to the paper's core argument. Simply, the author preferred to emphasize stochasticity. As previously said, I believe the link between exploration capabilities and batch size warrants more investigation.

[1] Yun, Chulhee, Shashank Rajput, and Suvrit Sra. "Minibatch vs local SGD with shuffling: Tight convergence bounds and beyond." arXiv preprint arXiv:2110.10342 (2021).

[2] Ahn, Kwangjun, Chulhee Yun, and Suvrit Sra. "SGD with shuffling: optimal rates without component convexity and large epoch requirements." Advances in Neural Information Processing Systems 33 (2020): 17526-17535.

[3] Yun, Chulhee, Suvrit Sra, and Ali Jadbabaie. "Open Problem: Can Single-Shuffle SGD be Better than Reshuffling SGD and GD?." Conference on Learning Theory. PMLR, 2021.

**Summary Of The Paper:**

### Summary

&nbsp;&nbsp; This research examines the effect of batch size on performance in multi-step reinforcement learning. In supervised-learning experiments, several researchers have found that the variance of neural networks during training has a significant effect on learnability and convergence, since the neural networks' stochasticity pushes them to global optimality regions. Typically, non-linearity properties of networks, such as activation functions and skip-connection, direct a large number of inflection points to loss surfaces with respect to parameter spaces. Various researchers have conducted technical and empirical studies to demonstrate loss surface of neural networks and escape of local optimal points by utilizing stochasticity of learning.

In contrast, in deep reinforcement learning, there are several hyper-parameters, hence the influence of batch size is rarely investigated. In addition, the multi-step approach that assesses value-functions as the sum of those on the last visited states has been extensively employed for empirical performance enhancement. However, there was no actual research demonstrating how these characteristics impact agents' performance. In this context, this work analyzes the impact of batch size and the number of steps of value-function on both online and offline reinforcement learnings. This study asserts that increasing variances by decreasing batch size enhances overall performance, and the bigger the number of steps that are likewise predicted to have a high variance does not ensure excellent performance.

In order to make a fair comparison, this research presented the Dopamine framework [1] that built the ALE framework and conducted extensive experiments to substantiate their hypotheses. There are a few results that do not accord with their claims, but in general, as batch size drops, network performance and variances of loss and gradient values rise. Instead, there are no global trends for the amount of steps, but there is a sweet spot for performance. For instance, the majority of studies stated that three or five is the optimal number of multi-steps like Dopamine framework.

[1] Machado, Marlos C., et al. "Revisiting the arcade learning environment: Evaluation protocols and open problems for general agents." Journal of Artificial Intelligence Research 61 (2018): 523-562.

**Summary Of The Review:**

Overall, the paper marginally show perfirmance enhancement using a batch-size and the number of steps in multi-step reinforcement learnings. In order to provide more general insights, this should be addressed by either improved theoretical statement or extensive experimental results across various environments.

---

> ### Author Response · Authors · 2022-11-11
> **Author response 1**
>
> We thank the reviewer for their feedback. It seems the main concern was with the clarity of the presentation of the phenomenon, as the reviewer was “unable of digesting a high-level summary of this research”, and felt that the “document appears to lack a logical structure”.
>
> As stated in the abstract, the main takeaway is that, surprisingly, the variance increase resulting from reduced batch size coupled with multi-step updates can be beneficial to learning. The structure of the paper, as indicated by the section titles, is as follows:
> 1. We pose the question: What is the role of batch sizes in value-based deep RL? (Introduction)
> 1. We investigate this by varying batch-size on a number of value-based agents trained on the ALE, observe that reduced batch size can help improve performance, and uncover under what circumstances reduced batch size helps (Section 2)
> 1. Having discovered this phenomenon, dig into what may be causing it (Section 3)
> 1. Investigate whether the phenomenon is present in other training regimes (Section 4)
>
> We have also added a more thorough background section, including an explanation of both, in Appendix A.
> Regarding state-of-the-art on the ALE, our intent was to present the phenomenon across a broad set of agents and training regimes. Our submission provides results on **20 different ALE games** for **four value-based online agents** (DQN, Rainbow, QR-DQN, IQN), **three sample-efficient agents** (DER, DrQ(eps), SPR), **two offline agents** (CQL, CQL+DR3), and using **4 different batch sizes** & **2-5 different n-step updates**. Following reviewer suggestions, we are also running more experiments on DM-Control, ProcGen, and are generating extra analyses for more games.
>
>
> We provide responses to some of the more specific concerns raised:
>
>
> > 1) I disagreed with the author's statement that multi-step reinforcement learning should be examined independently
>
> We do not feel we have made this claim. Could the reviewer point us to where it is made?
>
> > 2) At this time, I was unable to understand why this work manipulates and reveals the effect of multi-levels and its impact on batch-size by up to 15.
>
> Figure 1 hints at a performance improvement when reducing batch size for agents with multi-step returns, while Figure 2 confirms that performance boost is a consequence of combining reduced batch size with multi-step returns. Given this surprising finding, the intent of Figures 3 and 6 was to explore more extreme values of these two components so as to obtain a clearer picture of their ultimate effect on performance. Given that agents trained in low-data regimes (100K) use n-steps of 10, we chose to explore further.
>
> > 3)  At least at this stage, I believe the authors should provide a high-level statement regarding batch size to grasp the stochasticity of learners rather than give basic experiments to prove their hypothesis.
>
> We thank the reviewer for pointing us to these papers. However, the results there are for supervised learning settings (as the reviewer acknowledged), and only apply under specific assumptions about the supervised learning algorithms used. As such, they do not apply in the scenario we consider here. The experiments we conducted align well with standard setups for evaluating deep reinforcement learning algorithms.
>
> > 4)  there is no investigation of the exploration impact. To validate your hypothesis, you must offer consistent experimental results with modifying these hyper-parameters related to exploration
>
> Although we agree exploration plays a major role in any reinforcement learning agent, we feel it is orthogonal to our findings. Indeed, most ALE agents use an epsilon-greedy exploration strategy that starts at epsilon=1.0 (random action selection) and decays to a value of 0.01 over 250K agent steps. Nevertheless, we are running some extra experiments varying the target epsilon value to ensure the uncovered phenomenon is unaffected. Although the agents have not completed training, the results suggest decreasing $\epsilon$ seems to be more beneficial. It is interesting to observe that the variance double-down phenomenon remains present despite varying $\epsilon$ values. We have added these results to Appendix H, and included a discussion in Section 6.

---

> > ### Comment · Reviewer_74JP · 2022-11-17
> > **Response to author's response**
> >
> > Thank you for the detailed rebuttal. As comments by authors, my main concern is that it is unclear what logical points this paper want to deliver in ICLR2023. When I read authors responses, some of concerns regarding experimental results are resolved, however my problems with the logical structure of this paper remain.
> >
> > ### High-level concerns of this paper
> >
> > - In deep RL settings, there are a number of hyper-parameters that affect variances of Q-values or performance, including exploration strategy, frequency of updating networks (both actor and critic agent), learning rates, step sizes as well as batch-size. However, this paper supposed that the previous work [1] already determined  the best hyper-parameters in the ALE, hence the authors regarded that they did not need to discuss those parameters excluding batch-size and the number of steps. If the authors had proposed state-of-art algorithms, this empirical performance would be worth of sharing with public. However, in this paper, the authors simply asserted that manipulation of variances of Q-values during training should be key factor for improvement by displaying experimental results with varying batch-sizes. If the authors wish to experimentally insist this statement, I believe that they should present comparable experiments that take into account many hyper-parameters. Otherwise, at least, they must propose a conceptual theorem for high-level understanding of variance of Q-values in RL. Specifically, I disagree with this experimental reasoning for using random noises during training. In the prospective of variances, it is a pretty straightforward experiment, however it does not reflect all of phenomena regarding variances during training.
> >
> > - Regarding the ALE environment, I do not concur with the author's position. The authors assumed that variance increasing phenomena on ALE are worthy of presentation in ICLR2023, however, the ALE environments are only one of the RL environments and cannot represent the entire RL environment. I believe that in order to generalize the authors' claim, they must cover consistent phenomena in similar environments, such as Procgen and Mojoco using visual input. In the response, the authors have prepared of additional results, but I feel that the current revision lacks of experiments to persuade.
> >
> > - Otherwise, if the authors believe that investigation on the ALE environments are worth, they clearly specify where this phenomena distinctly happens among various ALE environments. There are various type of environments including sparse-reward environments. For example, Momentum revenge, Skii, Venture games requires exploration capability to find optimal policy, and several papers reported that variances or hyper-parameters influence agent performance during training. Hence, they must analyze this phenomena according to each environment of the ALE environment.
> >
> > ### Minor points of this paper
> >
> > **Entropy of experience replay buffer** : When I wrote a review, I would like to provide an useful empirical option to support your statement like manipulation of batch-size. I hypothesized that diversity on experience buffer can be one of elements, controlling variance of Q-values, and agent performance during training. Hence, I believed that the entropy value of experience buffer can be regarded as a metric of diversity. If the authors disagree this point, they do not need to take no notice of this points.
> >
> > - Reference.
> >
> > [1] Castro, Pablo Samuel, et al. "Dopamine: A research framework for deep reinforcement learning." arXiv preprint arXiv:1812.06110 (2018).

---

> > > ### Author Response · Authors · 2022-11-17
> > > **Response to high-level concerns**
> > >
> > > We thank the reviewer for their response. Before responding directly to the points raised, we would like to clarify a high-level point that we feel is at the heart of some of the concerns.
> > >
> > > The argument we are making in the paper is that **reduced batch size coupled with a longer update horizon can lead to improved performance, and this is unexpected given the existing literature**; we are _not_ claiming that "increasing variance improves performance". The variance analyses we performed are in service of our paper's argument, as a possible explanation for the observed phenomenon.
> > >
> > > We are wondering if our choice of name (for both the paper and the phenomenon) is causing some confusion by placing undue emphasis on the variance. If so, we would be open to renaming it to something else, such as "The small batch phenomenon in multi-step deep RL".
> > >
> > > Specific responses to each of your high-level concerns below:
> > >
> > > > In deep RL settings, there are a number of hyper-parameters…
> > >
> > > We are not claiming [1] determined the best hyper-parameters for each agent, but they do provide a reasonable baseline (on which many other papers base themselves on). Further, we are not claiming to provide a SoTA agent, but rather highlighting the surprising finding of combining reduced batch-size with multi-step updates, which in the vast majority of cases results in performance gains. To provide convincing evidence of this we ran approximately 26,361 separate experiments over a variety of agents, games, training regimes, batch sizes and n-steps, which is already quite comprehensive. We agree it would be interesting to run these types of explorations on all the other available hyper-parameters, but it is somewhat outside the scope of the message of the paper (as clarified above); further, the computational expense of an exercise of that form is rather prohibitive.
> > > Nevertheless, we did run some extra experiments varying exploration rate (see Appendix H) and learning rate (Appendix I).
> > >
> > > Regarding the request for a theorem, as you are likely aware, RL theory is limited to linear function approximators. However, appendix F and point #5 in our overview response to all reviewers strongly suggests the phenomenon is a purely deep phenomenon.
> > >
> > > > Regarding the ALE environment, I do not concur with the author's position…
> > >
> > > It is worth clarifying that the ALE is _not_ an environment, but a suite of 60 separate games of varying characteristics. We have run all our experiments on 20 of these games (as suggested by [2] (a paper published at ICML). There have been numerous past papers published in ICLR/ICML/NeurIPS that have provided scientific advances by focusing on the ALE (e.g. [3], [4], [5], just to give some ICLR-specific examples). Thus, we do not agree that having focused on the ALE is grounds for dismissal of our work.
> > >
> > > Nevertheless, prompted by your initial review, we have provided results with Rainbow on the ProcGen suite (16 separate environments, 4 batch sizes, 4 n-step values, and 3 seeds, totalling 768 separate runs) and MPO on DM-Control (12 separate environments, 5 batch sizes, 2 n-step values, and 10 seeds, totalling 1200 separate runs) in Appendix G. This was a total of  1200 separate runs for the DM-Control experiments. These results suggest the observed phenomenon is not limited to Atari games.
> > >
> > > Although the reviewer acknowledges that we have run extra experiments, they still “feel that the current revision lacks of experiments to persuade.” Could the reviewer please let us know what would constitute sufficient evidence for these extra environments, beyond the 1968 separate runs already provided?
> > >
> > > > Otherwise, if the authors believe that investigation on the ALE environments are worth…
> > >
> > > We agree a discussion of per-game phenomena could be useful. Although we are already providing some per-game results in the Appendix (see Figures 19 and 21), we can provide them for the other experiments run and add a discussion in the final version of the paper.
> > >
> > > > Entropy of experience replay buffer
> > >
> > > We are still unsure what the reviewer means by this. Entropy is defined as $-\sum_{x\in\mathcal{X}} p(x)\log p(x)$ for a set $\mathcal{X}$. Could the reviewer clarify what they have in mind as $\mathcal{X}$ and $p(x)$ with regards to the replay buffer?

---

> > > > ### Author Response · Authors · 2022-11-17
> > > > **Response to high-level concerns (references)**
> > > >
> > > > [1] Castro, Pablo Samuel, et al. "Dopamine: A research framework for deep reinforcement learning." arXiv preprint arXiv:1812.06110 (2018).
> > > >
> > > > [2] William Fedus, Prajit Ramachandran, Rishabh Agarwal, Yoshua Bengio, Hugo Larochelle, Mark Rowland, and Will Dabney. Revisiting fundamentals of experience replay. In International Conference on Machine Learning, pp. 3061–3071. PMLR, 2020.
> > > >
> > > > [3] Meire Fortunato, Mohammad Gheshlaghi Azar, Bilal Piot, Jacob Menick, Ian Osband, Alexander Graves, Vlad Mnih, Remi Munos, Demis Hassabis, Olivier Pietquin, Charles Blundell, and Shane Legg. Noisy networks for exploration. In Proceedings of the International Conference on Representation Learning (ICLR 2018), Vancouver (Canada), 2018.
> > > >
> > > > [4] Tom Schaul, John Quan, Ioannis Antonoglou, and David Silver. Prioritized experience replay. In Proceedings of ICLR, 2016, 2016.
> > > >
> > > > [5] Łukasz Kaiser, Mohammad Babaeizadeh, Piotr Miłos, Błazej Osi ˙ nski, Roy H Campbell, Konrad ´Czechowski, Dumitru Erhan, Chelsea Finn, Piotr Kozakowski, Sergey Levine, Afroz Mohiuddin, Ryan Sepassi, George Tucker, and Henryk Michalewski. Model based reinforcement learning for Atari. In International Conference on Learning Representations, 2020.

---

> > > > ### Comment · Reviewer_74JP · 2022-11-19
> > > > **Response to authors 2nd comments**
> > > >
> > > > Thank you for your feedback to my review.
> > > >
> > > > First of all, I leave a my comment on your inquiry regarding to high-level understanding of this paper. When I initially reviewed this paper, I had comprehensive knowledge of its contents and experimental findings. There is counter-intuitive phenomena wherein smaller batch-size increases performance in visual-inputs reinforcement learning. Specifically, the authors expanded the logical storyline of your paper (both initial draft and revised version) by empirically pointing out variance of agent components during training. Nonetheless, I contend that the authors lack a structural framework to describe this phenomena.
> > > >
> > > > If you wish to assert that lowering batch size influences on performance, you must explain why this batch-size effectively governs agents performance than other exploration, step-size and update frequency. In this sense, the authors completely rely on empirical finding with several ablation experiential results like Appendix H. In this regard, I believe that the authors have inadequately described the structural reasoning of this phenomena. In the top-tiered conferences like ICLR, the authors need to sufficiently describe this phenomena with technical frameworks and numerous experimental results with similar tendency the previous works did not address. In terms of experiments, I thought that the authors put numerous efforts to display empirically, but the authors must reveal the reason why this phenomena happens, and how batch-size and longer update horizon affects on agents performance in detail.
> > > >
> > > > ### Experiments and environments
> > > >
> > > > I appreciate the authors' enormous efforts in the rebuttal. I praise your diligence in actively addressing our concerns. However, without any technical framework for these experiments, it does not look like a worth of sharing this fact in top-tiered conferences. In this point, I believe that providing agent behaviors on each game makes this paper more valuable and increase readability and leads readers interest. Especially, by using this analysis, I thought that the authors enable to make a framework to understand this phenomena. However, in the rebuttal phase, the authors did not provide game-specific results in terms of batch-size and training horizons due to time constraints.
> > > >
> > > > In other aspects, experiments on other environments such as Procgen would be good examples because these environments have different properties than the ALE suite. The ALE suites have the set of deterministic environments, but since in ProcGen, one can make stochastic environments by controlling hyper-parameters, the displaying experimental results on several stochastic environments would be more helpful to understand variance effect of agents during training. However, in this rebuttal, the authors simply showed surrogated experimental results like Figure 35.
> > > >
> > > > ### Diversity of samples using small batch-size
> > > >
> > > > In my initial review, I have pointed to give an informative comment on effect of small episodic buffer by using diversity measure. So, as I mentioned in the previous response, if the authors analyze diversity of samples during training, it would be providing a technical framework to understand this phenomena. In continual learning studies, many paper have reported that sample diversity in the replay buffer has quite impact on algorithm performance in online-learning [1, 2]. Additionally, although we cannot directly figure out entropy value of samples, some of works approximate this value [3]. This approach have been actively used in the contrastive learning studies.
> > > >
> > > > To sum up, I thought that this work would establish the new axis of reinforcement learning. However, to present this paper in the top-tier AI conference, it requires to characterize a technical framework to describe this phenomena. I feel that the problem definition, logical approach and motivation in this paper are insufficient to explain the phenomena. In addition, I cannot enhance my rating based on the authors' assurance that experimental data would be included in the camera-ready version.
> > > >
> > > > ### Reference
> > > >
> > > > [1] Prabhu, Ameya, Philip HS Torr, and Puneet K. Dokania. "Gdumb: A simple approach that questions our progress in continual learning." European conference on computer vision. Springer, Cham, 2020.
> > > >
> > > > [2] Bang, Jihwan, et al. "Rainbow memory: Continual learning with a memory of diverse samples." Proceedings of the IEEE/CVF Conference on Computer Vision and Pattern Recognition. 2021.
> > > >
> > > > [3] Belghazi, Mohamed Ishmael, et al. "Mutual information neural estimation." International conference on machine learning. PMLR, 2018.

---

> ### Author Response · Authors · 2022-11-11
> **Author response 2**
>
> > 5)  The experimental setting was inconsistent in this work. In Figure 1, DQN, Rainbow, and IQN were employed as baselines, but in Figure 2, just DQN and Rainbow were used for the ablation investigation.
>
> The intent of Figure 1 was to showcase the initial experiment of varying batch size for four different agents. Figure 2 sets out to discover what component of Rainbow enables the performance boost with reduced batch-size. Here we are following the experimental analysis used by [1] and [2] to selectively add/remove components from DQN/Rainbow. The result is that observed phenomenon is a consequence of both reduced batch size and multi-step updates.
> We had already performed a similar analysis for the other two agents in Figures 9-13 in Appendix B, which show the same result. A clearer pointer to these additional plots is now in the main paper.
>
> > 6) I do not understand why the mini-batch split and random noise to target-value experiment results were reported. I felt that these trials were not really relevant to the paper's core argument.
>
> **Mini-batch split experiments:** Like most modern RL agents, we use the Adam optimizer, which adapts learning rates throughout training, where adjustments are made based on estimates computed from the sampled batches. We hypothesized that smaller batch sizes could have an effect on these estimates. The mini-batch split experiment was designed to test this hypothesis; our results suggest that this second-order effect is not the cause of the variance double-down phenomenon.
>
> **Random noise to target-value:** Our results demonstrate that reduced batch-sizes and multi-step learning results in improved performance (Figures 1-3), but also in increased variance in the losses and gradients (Figure 4). However, these two findings on their own are not sufficient to assert that “doubling down on variance” results in improved performance. To test this more directly one can explicitly and directly increase the variance in the loss; a natural way to do this  is by adding noise to the target values. Our results (Figure 5 (right) and Appendix B.2) suggest that, indeed, the increased variance plays a crucial role in the performance gains.
>
>
> > 7) If the authors wish to assert that a small number of batches increases performance by increasing stochasticity, they should examine more environment like Mujoco. If not, at least, the ALE-like Procgen should be chosen to support this notion. As just one setting employs ALE, its utilization is marginal.
>
> Although the focus of our paper was value-based agents on discrete control environments like the ALE, we agree it is worth exploring continuous control. As per the reviewer suggestion, we are currently running experiments on DM-Control [3] (similar to MuJoCo) with MPO [4] and on Procgen with Rainbow. In DM-Control, the phenomenon is clearly visible with n-step=5: reduced batch size results in improved performance (see Appendix G for these results). Due to computational constraints, the remaining experiments with these extra environments will take a little more time, but we will add them to the paper (and report here) when they are done.
>
>
> > 8) Simply said, adjusting mini-batches has a minor effect on performance, however not all algorithms adhere to this hypothesis.
>
> Could the reviewer clarify what is meant by “not all algorithms adhere to this hypothesis?
>
> > 9) Otherwise, they should at least characterize the entropy of of batches for learners during training.
>
> Could the reviewer clarify what is meant by “entropy of batches for learners”?
>
> > 10) I believe that it did not meet the ICLR standard for reproducing experimental results.
>
> The majority of our experiments were simple one-line modifications of the gin-config files used by Dopamine (e.g. changing batch size for DQN involves simply changing [this line](https://github.com/google/dopamine/blob/master/dopamine/jax/agents/dqn/configs/dqn.gin#L37)). The only exceptions to this are the results of Figures 4 and 5; since these results are more analytical in nature we felt it less critical to include the code. Nevertheless, they are simple modifications to the Dopamine code, so we have included the necessary modifications as part of the supplemental material.
>
> ### References:
> 1. Hessel, Modayil, Hasselt, Schaul, Ostrovski, Dabney, Horgan, Piot, Azar, and Silver.  Rainbow: Combining Improvements in Deep Reinforcement Learning.  AAAI 2018
> 1. Johan Samir Obando Ceron and Pablo Samuel Castro.   Revisiting rainbow:  Promoting more in-sightful and inclusive deep reinforcement learning research.  ICML 2021.
> 1. Yuval Tassa, Yotam Doron, Alistair Muldal, Tom Erez, Yazhe Li, Diego de Las Casas, David Bud-den,  Abbas Abdolmaleki,  Josh Merel,  Andrew Lefrancq,  Timothy P. Lillicrap,  and Martin A.Riedmiller. Deepmind control suite. arXiv, 2018
> 1. Abbas Abdolmaleki, Jost Tobias Springenberg, Yuval Tassa, Remi Munos, Nicolas Heess, and Mar-tin Riedmiller. Maximum a posteriori policy optimisation. ICLR2018

---

### Official Review · Reviewer_fBLk · 2022-11-03

**Confidence:** 4
**Clarity, Quality, Novelty And Reproducibility:** The paper is very well-written and ve…
**Correctness:** 2
**Technical Novelty And Significance:** 2
**Empirical Novelty And Significance:** 2
**Recommendation:** 3

**Strength And Weaknesses:**

+ The paper observes that a larger variance caused by using a smaller batch-size leads to higher returns for multi-step objectives in RL. This is a phenomenon that jives well with existing empirical results, e.g,. the fact that on-policy methods also have a huge variance in their TD loss but do seem to achieve a good return. So the effect pointed in this paper is perhaps real.
+ The paper has extensive experiments whether this phenomenon is checked for the low-data regime, for offline learning,

-- All experiments were conducted over 3 seeds, how large are the error bars for more seeds?
-- For example, in Fig 2 left how do we control for the fact that the multi-step objective is quite different and therefore other parameters such as learning rate and weight decay also determine how well the function is learned. In this sense, I think the main premise of this study is misguided. A more reasonable hypothesis would be: for the best value of other hyper-parameters, do training runs with smaller batch-size and multi-step objective work better?
-- “the performance boost is correlated with increased variance on both these fronts; we are dubbing this the variance double-down phenomenon” I don’t think there is enough evidence yet in this paper to ascribe to the improved returns (esp. since the error bars are quite large in Fig. 4 for 3 seeds) to improved variance. It is understood that this is a non-convex objective and therefore it is difficult to make such a claim rigorously.
-- “It is thus possible that smaller batch sizes have a second-order effect on the learning-rate adaptation that benefits agent performance” This sentence makes sense, can the authors elaborate?
-- The results on low-data and offline regimes seem to throw a spanner in the main hypothesis; they are quite inconclusive.
-- “One might argue that reducing the batch size without additional training effectively mitigates overestimation, simply because each transition is trained on fewer times” — there does not seem to be any evidence in  the paper to make this claim.


**Summary Of The Paper:**

This paper shows that the default batch-size of 32 is not optimal for typical RL algorithms (DQN, Rainbow, QR-DQN, IQN) and that decreasing the batch-size, or also increasing the batch-size, can give better human-normalized returns for some algorithms.

**Summary Of The Review:**

This paper investigates a phenomenon where the increased variance due to smaller batch-sizes seems to lead to improved performance for RL methods. This is an experimental paper but there does not seem to be enough evidence for this claim either way. The authors should do more controlled experiments to study this phenomenon more systematically.

---

> ### Author Response · Authors · 2022-11-11
> **Author response**
>
> We thank the reviewer for their useful suggestions. We are glad the reviewer found our paper **well written, clear, and original**. If we are reading the review correctly, the main concern is that we might not be providing enough evidence for the variance double-down phenomenon (namely, that the variance increase resulting from reduced batch size coupled with multi-step updates can be beneficial to learning).
>
> We conducted our investigation on **20 ALE games**, with **nine different agents** (DQN, Rainbow, QR-DQN, IQN, DER, DrQ(eps), SPR, CQL, and CQL+DR3), on **three different training regimes** (Online, Low data regime, and Offline), and using **4 different batch sizes & 2-5 different n-step updates**. Following reviewer suggestions, we are also running more experiments on DM-Control, ProcGen, and are generating extra analyses for more games.
> Regarding the choice of seeds, we used the number of seeds suggested by [1, 2, 3]: 3 seeds for Atari 100M, 6 seeds for Atari 100k and 5 seeds for the offline experiments. Further, we are aggregating across tasks and using robust statistical measures, as suggested by [1].
>
> We include more specific answers to issues raised below:
>
> > 1) in Fig 2 left how do we control for the fact that the multi-step objective is quite different and therefore other parameters such as learning rate and weight decay also determine how well the function is learned.
>
> For Figure 2, we followed the approach taken by [3], which added/removed components while keeping all other hyper-parameters fixed. Nonetheless, we are running some extra experiments varying the learning rate to evaluate its impact on update horizon, as suggested. Although these experiments are still running, we include partial results in Appendix I, which suggest we are in fact using the optimal learning rate. We will update the results in the paper once these experiments have completed.
>
> > 2) I don’t think there is enough evidence yet in this paper to ascribe to the improved returns (esp. since the error bars are quite large in Fig. 4 for 3 seeds)
>
> The plots in Figure 4 are for a single game (Asterix), which is why the confidence intervals are larger. However, it is worth noting that in most figures, the confidence intervals do not overlap, which indicates a statistically significant difference. Nonetheless, we agree that more evidence is also welcome, so we have run these same analyses on four additional games (Breakout, Pong, MsPacman, and SpaceInvaders) and included them in Appendix C.1. The results with these extra games are consistent with what we found with Asterix.
>
> > 3) The results on low-data and offline regimes seem to throw a spanner in the main hypothesis; they are quite inconclusive.
>
> In general, we have found the 100K regime to be quite noisy (a fact also raised by [1]). Thus, we extended training to 30M steps (Figure 6 bottom) and, indeed, found the variance double-down phenomenon clearly present.
>
> In the offline setting, the double down phenomenon is very clear in DR3+CQL and DQN. We agree that the phenomenon is only mildly present in CQL. We hypothesize that it may be a consequence of the  implicit under-parameterization exhibited by this algorithm (and which DR3 aimed to mitigate).
>
> > 4) “One might argue that reducing the batch size without additional training effectively mitigates overestimation, simply because each transition is trained on fewer times” —  there does not seem to be any evidence in the paper to make this claim.
>
> Indeed, this is precisely what we say in the following sentence: “A closer look at the learning curves (Fig 25 in the appendix) suggests reduced overfitting is not the main factor explaining our results.” To elaborate a bit more: if reduced overfitting were the cause of performance improvements, we would see an improvement with reduced batch-size and n-step=1 (top row).
>
> ### References
> 1. Rishabh Agarwal, Max Schwarzer, Pablo Samuel Castro, Aaron Courville, and Marc G Bellemare.Deep reinforcement learning at the edge of the statistical precipice. InThirty-Fifth Conference onNeural Information Processing Systems, 2021.
> 1. Aviral Kumar, Rishabh Agarwal, Tengyu Ma, Aaron Courville, George Tucker, and Sergey Levine.Dr3:  Value-based deep reinforcement learning requires explicit regularization.  InInternationalConference on Learning Representations, 2021
> 1. Matteo Hessel, Joseph Modayil, Hado van Hasselt, Tom Schaul, Georg Ostrovski, Will Dabney, DanHorgan, Bilal Piot, Mohammad Azar, and David Silver.  Rainbow: Combining Improvements in Deep Reinforcement Learning.  InProceedings of the AAAI Conference on Artificial Intelligence, 2018

---

### Author Response · Authors · 2022-11-16
**Overview of responses and new experiments**

We would like to summarize the changes we have made to our submission in response to the reviewer concerns:

1. **Extra environments:** We have run additional experiments with MPO on DM-Control (similar to MuJoCo) and Rainbow on ProcGen. Although not as comprehensive as the ALE experiments already included in the paper, we find the variance double-down phenomenon _is_ still present in these domains. See Appendix G.
1. **Extra games for variance analysis:** Reviewer fBLk questioned whether there was enough evidence for ascribing improved returns to increased variance. To address this, we have run the same analyses on four extra games (Breakout, Pong, MsPacman, and SpaceInvaders), which show consistent results. See Appendix C.1.
1. **Different learning rates:** In Appendix I we evaluated QR-DQN with varying learning rates, as suggested by fBLk. Overall it does seem like larger batch sizes benefit from reduced learning rates (left column in Figure 37). Nonetheless, _under the default learning rate settings_ (middle column of Figure 37), reduced batch sizes are able to match their performance. When considered with the wall-time savings argued in the Discussion (Figure 8), this is a net-positive for opting for reduced batch sizes.
1. **Varying exploration rates:** Reviewers 74JP and iTrT suggested we should investigate the effect on exploration. To do so, we varying the target $\epsilon$-exploration rate used by all the ALE agents, and found that the variance double-down phenomenon remains present despite varying $\epsilon$ values. See Appendix H.
1.  **Experiments with linear function approximation:** Reviewer iTrT questioned the validity of restricting ourselves to an empirical evaluation with deep networks. In Appendix F we ran experiments with linear function approximators on three classic control tasks; the results there suggest that _the variance double-down phenomenon is a purely deep network phenomenon_.

As requested by 74JP, we have also included the source code used for our experiments in the supplemental.

For your convenience, the udpated version of our submission uses blue text to indicate changes we have made since initial submission.

We feel we have addressed all other concerns to each corresponding reviewer, below. Please let us know if there is something you feel is still not properly addressed and we would be happy to respond. Otherwise, please consider revising your score.

Thank you once again for your time!

---

### Author Response · Authors · 2022-11-18
**Final day for discussion with authors**

Dear reviewers, today is the last day that authors are allowed to participate in discussions. We believe we have addressed all your concerns, and have run all the extra experiments suggested.

So far only reviewer 74JP has responded to our rebuttal. Please let us know if there are still questions/concerns we can address; if not, please consider revising your score.

Thank you all once again for your efforts!

---

### Decision · Program_Chairs · 2023-01-20

**Decision:**

Reject

**Justification For Why Not Higher Score:**

Issues with experimental design and analysis of the central phenomena studied in the paper.

**Justification For Why Not Lower Score:**

N/A

**Metareview: Summary, Strengths And Weaknesses:**

The paper studies the effect of batch size in multistep RL. The central finding in the paper is that reducing batch size and increasing number of steps for TD learning simultaneously can be helpful in many cases. The somewhat surprising finding is that both changes increase the variance of the loss and gradients. There was a rich discussion around this work. On the positive side, reviewers appreciated the potential impact of the study and the extra experiments done during the rebuttal stage. However, there are some pending concerns with the current work. First, the experimental protocol of keeping other hyperparameters fixed while varying batch size fixed is strange. As the reviewers pointed out, varying of learning rates and exploration hyperparameters should be in conjunction with batch size for best performance, as they affect the variance as well. In this regard, the new results e.g., Figure 37 cast somewhat of a shadow on how meaningful is the main claim of the paper, as tuning other params for higher batch sizes can also improve their performance. Second, the somewhat mixed performance in the offline setting was another concern whether the claimed observation about batch sizes needs further assumptions / restricted to certain environments and algorithms -- amongst the 3 algorithms considered, CQL is the most common in practice and the claimed trend does not hold there. Finally, some of the reviewers felt that even if the trend was highly robust (which it does not seem given the above points), a more detailed understanding of the phenomena would improve the work. I encourage the authors to focus on this feedback for a future submission.